# Historical reconstruction of background air pollution over France for 2000-2015

Elsa Real[1], Florian Couvidat[1], Anthony Ung[1], Laure Malherbe[1], Blandine Raux[1], Alicia Gressent[1], Augustin Colette[1]

[1]INERIS, France

*Correspondence to*: Elsa Real (elsa.real@ineris.fr)

**Abstract.**

This paper describes a 16-year data set of air pollution concentrations and air quality indicators over France. Using a kriging method that combines background air quality measurements and modelling with the CHIMERE Chemistry Transport Model,

hourly concentrations of $NO_2$, $O_3$, $PM_{10}$ and $PM_{2.5}$ are produced with a spatial resolution of about 4 kilometers. Regulatory indicators (annual average, SOMO35, AOT40 etc…) are also calculated from these hourly data. The $NO_2$ and $O_3$ datasets cover the period 2000-2015, as well as the annual $PM_{10}$ data. Hourly $PM_{10}$ concentrations are not available from 2000 to 2007 due to known artefacts in PM10 measurements. $PM_{2.5}$ data are only available from 2009 onwards due to the limited number of measuring stations available before this date. The overall dataset was evaluated over all years by a cross-validation process

against background stations (rural, sub-urban and urban), to take into account the data fusion between measurement and models in the method. The results are very good for $PM_{10}$, $PM_{2.5}$ and $O_3$. They show an overestimation of $NO_2$ concentrations in rural area, while $NO_2$ background values in urban areas are well represented. Maps of the main indicators are presented over several years and trends are calculated. Finally, exposure and trends are calculated for the three main health-related indicators: annual averages of $PM_{2.5}$, $NO_2$ and SOMO35. The DOI link for the dataset is http://doi.org/10.5281/zenodo.5043645 (Real et al.,

2021). We hope that the publication of this open dataset will facilitate further studies on the impacts of air pollution.

## 1. Introduction

Air pollution is a major environmental risk for human health and ecosystems in Europe. Over the past decades the European Union (EU) has put in place several measures to reduce anthropogenic emissions of pollutants. In response to emissions

reductions, concentrations of $SO_2$, $NO_2$ and particles measured over Europe show a clear decrease since 1990 (EEA, 2018; EMEP, 2016).

The evolution of $O_3$ trends is less clear, despite the decrease in its precursors. The magnitude of high ozone episodes has decreased while annual average ozone levels measured at EMEP stations were increasing in the 1990s, and show a limited

negative trend from 2002. As shown in the Tropospheric Ozone Assessment Report (Tarasick et al., 2019), this feature is generally attributed to the changing global tropospheric ozone baseline for which further hemispheric control strategies are needed. The same conclusions could be drawn from the Malherbe et al. study, which focused on France, with significant reductions in $NO_2$ and particles concentrations and an increase in average $O_3$ offset by a slight decrease in peak $O_3$. Despite

these reductions in emissions and pollutant concentrations (with the exception of the annual average $O_3$), a proportion of French citizens is still exposed to concentrations above the EU limit and target value and air quality in EU remains  one of the main reasons for premature deaths (IHME, 2013).

As a complement to observations (which provide only partial spatial information), accurate, highly spatial resolution and up-
to-date air pollution maps are important information for assessing air pollution trends and exposure. They should provide geographically detailed information on the concentrations of air pollutants over the whole territory. These maps serve as a basis for informing citizens information, for designing and stratifying monitoring networks, for supporting policy strategies and measuring their impact. They are also used to estimate population exposure to air pollutants, which is essential for epidemiological studies.

On a European scale, different mapping approaches have been used to produce maps of pollutant concentrations. These maps can be obtained by modeling using a regional Chemistry Transport Model (CTM) that simulates the concentration of pollutants over Europe. However, these models cannot always be used over the whole Europe with a high resolution and have some biases and limitations in spatial representativeness. Regression methods (Briggs et al., 2000; Beelen et al., 2007) are also used at different scale. These stochastic modelling techniques develop statistical associations between potential 'predictor variables'
(land use, emission sources, topography) and measured pollutant concentrations in order to predict concentration at an unsampled site. Other frequently used techniques are kriging techniques. These geostatistical techniques are based on the assumption that the data are spatially autocorrelated, and therefore take into account the distances between measurements and the spatial structure of the variable. Different types of kriging are used to map the concentrations of air pollutant. Over France, kriging methods combining information from a regional CTM (CHIMERE, (Mailler et al., 2017)) and observations are
produced daily by the Prev'air operational forecasting and mapping System for Air Quality  (Rouïl et al., 2009). Since 2003 (for ozone), and 2005 for $PM_{10}$, the concentrations maps simulated for the day before in Prev'air are corrected each morning using observations. The kriging technique used in Prev'Air has evolved over time, and $PM_{2.5}$ and $NO_2$ concentrations are now also corrected for the day before. Today, a kriging of hourly observations with CHIMERE as external drift is applied to map $NO_2$ and $O_3$ concentrations. Since 2017, for the mapping of $PM_{10}$ and $PM_{2.5}$ concentrations, the method used is an hourly
cokriging of $PM_{10}$ and $PM_{2.5}$ data with CHIMERE as external drift. These choices are the results of successive studies that compared different kriging techniques (Malherbe and Ung, 2009, Beauchamp 2015a). A similar methodology was implemented for an earlier reconstruction of outdoor air pollution in Europe for the period 1989-2008 in (Bentayeb et al.,

2014). There are also ambient air pollution maps produced at European scale at 1km resolution by the European Environment Agency, but only for selected annual indicators and without consistency for multi-year reconstructions (Horálek et al., 2012, 2020). The Copernicus Atmosphere Monitoring Service has also produced European analyses since 2015, but again there is no multi-year consistency as these European maps are produced on an annual basis with gradually improving methodologies

(Marécal et al., 2015). At Global scale, the Global Burden of Disease also makes available air pollution exposure maps, a recent update of the methodology was presented in (Shaddick et al., 2017), but the resolution is 0.1 degrees or about 10km.

The purpose of this paper and the associated datasets is to present and provide mapped data of $O_3$, $NO_2$, $PM_{10}$ and $PM_{2.5}$ concentration at high spatial and temporal resolution and associated regulatory indicators covering the French metropolitan territory for the period 2000-2015 (2007-2015 and 2009-2015 for hourly concentrations of $PM_{10}$ and $PM_{2.5}$). The same kriging

technique as in the Prev'air system is used to combine modelled and observed concentrations. Hourly concentrations of $PM_{10}$, $PM_{2.5}$, $NO_2$ and $O_3$ are produced and mapped over France and these hourly data are then used to calculate and map European and French air quality standards.

## 2. Methods

Model outputs and measurements from the permanent monitoring network were combined by external drift kriging (Malherbe

and Ung, 2009; Benmerad et al., 2017) to construct hourly concentration maps over France for a long period: 2000 to 2015. Details on the input data and methods used are described in the following paragraphs. From these corrected hourly concentration data, annual regulatory air quality maps are then constructed over France.

### 2.1 Monitoring data

Hourly measurements are extracted from validated reference data sets. For France, observations are extracted from the national

air quality databases: BDQA (Base de Données de Qualité de l'Air) before 2013 and GEODAIR (https://www.lcsqa.org/fr/les-donnees-nationales-de-qualite-de-lair) after 2013; and from the Airbase database (https://www.eea.europa.eu/themes/air/air-quality/map/airbase) for other European countries from 2000 to 2012 and from AQ e-reporting (https://www.eea.europa.eu/data-and-maps/data/aqereporting-8/aq-ereporting-products) from 2013 to 2015. All background monitoring data over the spatial domain are used in the kriging procedure, except for stations with measurements above the 95

percentiles. This includes rural, suburban and urban stations but excludes industrial and traffic stations that are representative of very local concentration, difficult to reproduce in a national scale mapping system. The number of background monitoring sites for each type of station and for each year is summarize in Table 1.

**Table 1: Number of background French monitoring sites for the years 2000 to 2015**

| | 2000 | 2001 | 2002 | 2003 | 2004 | 2005 | 2006 | 2007 | 2008 | 2009 | 2010 | 2011 | 2012 | 2013 | 2014 | 2015 |
|---|---|---|---|---|---|---|---|---|---|---|---|---|---|---|---|---|
| $O_3$ | 284 | 310 | 337 | 362 | 378 | 396 | 404 | 405 | 399 | 385 | 376 | 360 | 347 | 318 | 319 | 331 |
| $NO_2$ | 274 | 290 | 299 | 322 | 337 | 353 | 353 | 350 | 352 | 337 | 334 | 316 | 299 | 284 | 282 | 300 |
| $PM_{10}$ | 119 | 125 | 171 | 212 | 219 | 238 | 126 | 219 | 252 | 241 | 249 | 245 | 240 | 218 | 173 | 251 |
| $PM_{2.5}$ | | | | | | | | | | 62 | 69 | 74 | 84 | 89 | 90 | 105 |

Until 1 January 2007, operational monitoring of $PM_{10}$ and $PM_{2.5}$ was carried out in France by automatic measuring systems of the TEOM ($PM_{10}$, $PM_{2.5}$) or Beta ($PM_{10}$) type. However, compared to the reference method EN 12341 (gravimetry), these systems underestimate the concentrations of particles. This is a known artefact related to the loss of semi-volatile compounds. To correct $PM_{10}$ measured concentrations measured before 2007, a simple approach consists in applying a uniform correcting factor over France. This method is not suitable for correcting hourly or daily concentrations, but it has been shown to work well for annual average $PM_{10}$ concentrations (Malherbe et al., 2017, Bessagnet et al., 2008). The factor (1.36) is a median value calculated on the $PM_{10}$ data from "reference" sites (Bessagnet et al., 2008). As a consequence, for the period 2000 to 2006, the only $PM_{10}$ indicator available is the annual average concentration. Concerning PM2.5, given the few reference measurements available before 2009, the reliability of even annual measurements is low. It was therefore decided to apply the kriging methodology only from the year 2009 onwards, for which the change in measurement method had become widespread.

## 2.2 CHIMERE simulations

The CHIMERE chemistry-transport model (Couvidat et al., 2018) is used to estimate air pollution levels for metropolitan France, with a resolution of about 4 km (0.06°×0.03°) over the year 2000 to 2015. This model has long been implemented and assessed in France as the main component of the national air quality forecasting and monitoring system PREV'AIR (Honoré et al., 2008). Two types of input data are used to simulate concentrations.

Prior to 2010, a configuration similar to the one use in the EURODELTA-Trends project (Colette et al., 2017) is used. The methodology of Colette et al. (2017) is used to reconstruct the emissions of main air pollutants (Non Methanic Volatile Organic Compound (NMVOC), NOx, CO, $SO_2$, $NH_3$, and Primary PM): the annual emissions of each country, broken down by SNAP (Selected Nomenclature for reporting of Air Pollutants) sectors, are estimated using the GAINS (Greenhouse gases and Air pollution Interactions and Synergies) model (Amann et al., 2011) for the years 2000, 2005, and 2010 . To derive emissions for intermediate years, sectorial results for 5-year periods are linearly interpolated. Meteorological data are simulated with the Weather Research and Forecast Model (WRF version 3.3.1; Skamarock et al., 2008) from 2000 to 2010.

For the period 2011 to 2015, year-to-year emissions of the main pollutants are taken from the Cooperative programme for monitoring and evaluation of long range transmission of air pollutants in Europe (EMEP) programme available

at http://www.emep.int. Annual meteorological data were provided by ECMWF with the Integrated Forecasting System (IFS) model with data assimilation.

For these two datasets, the spatialization of emissions over France is performed with a 1 km proxy based on the national bottom-up emission inventory (available at http://emissions-air.developpement-durable.gouv.fr/) which feeds the CHIMERE emission pre-processor described in Mailler et al. (2017). Furthermore, Denier van der Gon et al. (2015) showed that primary PM emissions from residential wood burning can be underestimated by up to a factor 2-3 over Europe because the emissions largely lack semi-volatile compounds. To compensate this underestimation, a country correction factor determined from Denier van der Gon et al. (2015) is applied over the whole period.

## 2.3 Kriging

Hourly atmospheric concentration fields are estimated by universal kriging, a geostatistical method. Kriging aims to estimate the value of a random variable (random process which describes the observations) at locations from the measurements. Kriging relies on the concept of spatial continuity which implies that measurements that are close to each other will be more similar than distant measurements. In addition, kriging requires a good knowledge of the spatial structure of the interpolation domain which is represented by the variogram or co-variogram (second order properties) of a random function (Goovaerts, 1997; Wackernagel, 2003; Chiles and Delfiner, 2012; Lichtenstern, 2013). Kriging involves deriving linear combination of the observations which ensures the minimal estimation variance under a non-bias condition. At a point $s_0$, the concentration estimate $\widehat{y(s_0)}$ is given by equation 1.

$$\widehat{y(s_0)} = \sum_{i=1}^{N} \lambda_i y(s_i)$$

**Equation 1**

Where $y(s_i)$, i=1…N, are the observed concentrations at sampling locations through the entire domain (unique neighborhood) or within a limited neighborhood of $s_0$ (moving neighborhood), and $\lambda_i$, i=1…N, are the kriging weights.

Among the kriging methods, the universal kriging (especially external drift kriging) allows to consider additional information to make estimate more accurate. This approach is based on a linear regression with auxiliary variables and a spatial correlation of the residuals and allows to combine simultaneously observations and additional information. The main hypothesis is that the global mean of the random variable is not constant through the domain and it relies on explanatory variables. This kriging technique has been used for several years in the monitoring air quality system for spatial interpolation at the regional scale (PREV'AIR, Malherbe et Ung, 2009). For $y(s_0)$, which is the pollutant concentration to be estimated at a location $s_0$, the hypothesis is a linear relation between $y(s_0)$ and the considered auxiliary variables as explained by equation 2 and 3.

$$y(s_0) = m(s_0) + \varepsilon(s_0)$$

**Equation 2**

$$m(s_0) = b_0 + b_1 x_1(s_0) + b_2 x_2(s_0) + \cdots + b_p x_p(s_0)$$

**Equation 3**

Where $m(s_0)$ is the drift of the mean, $b_0, b_1, \ldots, b_p$, are the coefficients of the linear regression, and $x_0, x_1, \ldots, x_p$, are the auxiliary variables. $\varepsilon$ corresponds to the stationary random process which is associated with a semi-variogram. In addition, the kriging weights must satisfy the drift condition described in equation 4.

$$\forall x_p : x_p(s_0) = \sum_{i=1}^{N} \lambda_i x_p(s_i)$$

**Equation 4**

In this work, kriging is performed with surface monitoring observations and the drift is described by the outputs from the CHIMERE chemistry transport model. European stations located outside the French domain are included in the kriging to increase accuracy at the borders. The kriging is performed using a moving neighbourhood as this allows for local adjustment of the relationship between the measurements and CHIMERE. The concentration at each grid point is estimated within a window of 80 monitoring sites. This number has been adjusted in previous studies by sensitivity tests (Benmerad et al., 2017;

Beauchamp et al., 2017). In addition, smoothing is applied to avoid discontinuities in the map (Beauchamp et al., 2015b); the smoothing methodology was adapted from Rivoirard and Romary (2011). The final output resolution is the same as for the CHIMERE model: approximately 4 km resolution (0.06°×0.03°).

For $PM_{10}$ (particles with a radius < 10 µm) and $PM_{2.5}$ (particles with a radius < 2.5 µm) a co-kriging with external drift is applied. Co-kriging is an extension of kriging to the multivariate case. It allows the estimate of $PM_{10}$ or $PM_{2.5}$ concentrations

by a linear combination of the two-variable data. The particularity of co-kriging is the use of the cross variance or semi-variance between the principal variable and the secondary variable. In the case of co-kriging with external drift, the simple and cross variograms are built based on residuals (Fouquet et al., 2007). Co-kriging allows to take into account the correlation between $PM_{10}$ and $PM_{2.5}$ and to improve consistency between $PM_{10}$ and $PM_{2.5}$ estimates (Beauchamp et al., 2015a). This cokriging also allows $PM_{2.5}$ estimate to benefit from the higher density of $PM_{10}$ monitoring stations.

### 2.4  Output: regulatory air quality indicators

From the hourly kriged concentrations, several air quality indicators (regulatory and used in health impact assessment) are calculated and mapped over France. The complete list and definition of these indicators are given in Table 2.

**Table 2: Yearly regulatory air quality indicators from EU legislation or French legislation and usual indicators.**

| ID | Pollutant | Statistics | Threshold | Threshold origin | Target to protect |
|---|---|---|---|---|---|
| NO2_avgannual | NO2 | Yearly average | 40 µg.m$^{-3}$ | Limit value (EU) | Human health |
| O3_avgannual | O3 | Yearly average | | | |
| O3_AOT40 | O3 | AOT40* from May to July | 6000 µg.m$^{-3}$ | Long-term objective | Vegetation |
| O3_AOT40_5years | O3 | AOT40* from May to July (5 years average) | 18000 µg.m$^{-3}$ | Target value (EU) | Vegetation |
| O3_SOMO35 | O3 | Sum of excess of max daily 8-hour averages over 35 ppb (= 70 µg m$^{-3}$) calculated for all days in a year; SOMO35 (Sum Of Means Over 35 ppb) | | Health Impact Assessment | Human health |
| O3_T120 | O3 | Number of days for which the running average over 8h exceeds 120 µg.m$^{-3}$ | | Quality objective (EU) | Human health |
| O3_T120_3years | O3 | Number of days for which the running 8h average exceeds 120 µg.m$^{-3}$ (averaged over 3 years) | Not to exceed more than 25 days a year | Target value (EU) | Human health |
| O3_T180 | O3 | Number of hours exceeding the average value of 180 µg.m$^{-3}$ | | Recommendation and Information Threshold (France) | Human health |
| O3_T240 | O3 | Number of hours exceeding the average value of 240 µg.m$^{-3}$ | | Alert threshold (France) | Human health |
| PM10_avgannual | PM10 | Yearly average | 40 µg.m$^{-3}$ | Limit value (EU) | Human health |
| PM10_t50 | PM10 | Number of days exceeding the average value of 50 µg.m$^{-3}$ | Not to exceed more than 35 days a year | Limit value (EU) | Human health |
| PM10_t80 | PM10 | Number of days exceeding the average value of 80 µg.m$^{-3}$ | | Alert threshold (France)) | Human health |
| PM25_avgannual | PM25 | Yearly average | 25 µg.m$^{-3}$ | Limit value (EU) | Human health |

*AOT 40 (expressed in µg / m³.hour) means the sum of differences between hourly concentrations greater than 80 µg / m³ (= 40 ppb or part per billion) and 80 µg / m³ for a given period using only the values 1 hour measured daily between 8 am and 8 pm.

### 3. Data validation

Usually the quality of the estimated concentrations maps is assessed using statistical indicators that compare observations and estimated concentrations at the monitoring stations in the domain. Here, information of all background stations inr the domain is already used to produce the maps. Therefore, for a fair comparison, the cross-validation method is used. The cross-validation method calculates the quality of the spatial interpolation for each measurement station point from all available information except the selected station point, i.e. it retains one data point and then makes a prediction at the spatial location of this point. This procedure is repeated for all measurement points in the available set, thus allowing the quality of the predicted values to be assessed at locations without measurements (provided they are within the area covered by the measurements).

It was noticed that the scores are systematically different on rural and urban stations (even though the kriging technique used here is not differentiate by the type of station). This is why, the results of the cross-validation are described by pollutant and differentiated by stations type (rural and urban types are presented here). Three statistical indicators are calculated on the basis of the daily average concentration: the mean bias, the root mean squared error (RMSE) and the pearson correlation ($r^2$). For each year, they are first calculated on the "left out" station and then the median values over all stations are calculated.

Leave-one-out validation is a commonly used method in the air quality community (see for example ETC reports on air quality mapping (ETC, 2020)) which is presently recommended by FAIRMODE (FAIRMODE guidance, 2020). However, scores derived from the results of the leave-one-out validation might be influenced by areas where the density of sampling points is highest. For this reason, during the FAIRMODE project (Riviere et al., 2019), for which a kriging method similar to the one conducted here was conducted, a comparison has been performed between cross-validation results obtained by the leave-one-out cross-validation and cross-validation results obtained by the 5-fold cross validation (leave-20%-station-out CV). Results and related scores were very similar. We therefore decided to keep to the leave-one-out cross-validation process for the validation of our kriging results.

### 3.1. PM₁₀

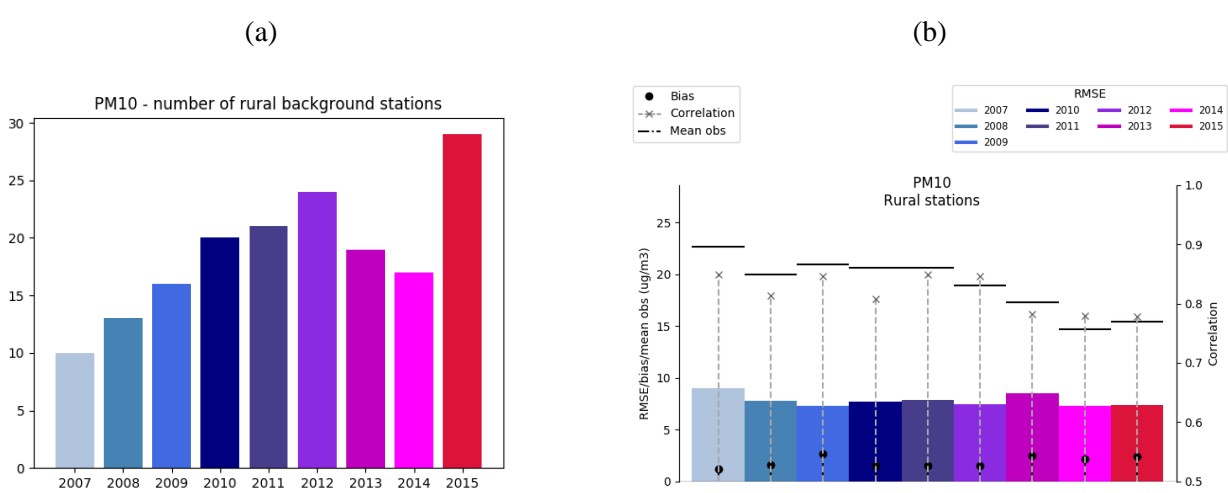

(a)                                                     (b)

Figure 1: PM₁₀: statistical indicators calculated using cross-validation technique on daily mean PM₁₀ values measured and estimated over RURAL background stations for the years 2007 to 2015. (a) number of rural stations for each year; (b) mean bias (black circles), RMSE (coloured rectangles), correlation (grey crosses and the associated dashed lines) and mean observation (horizontal lines).

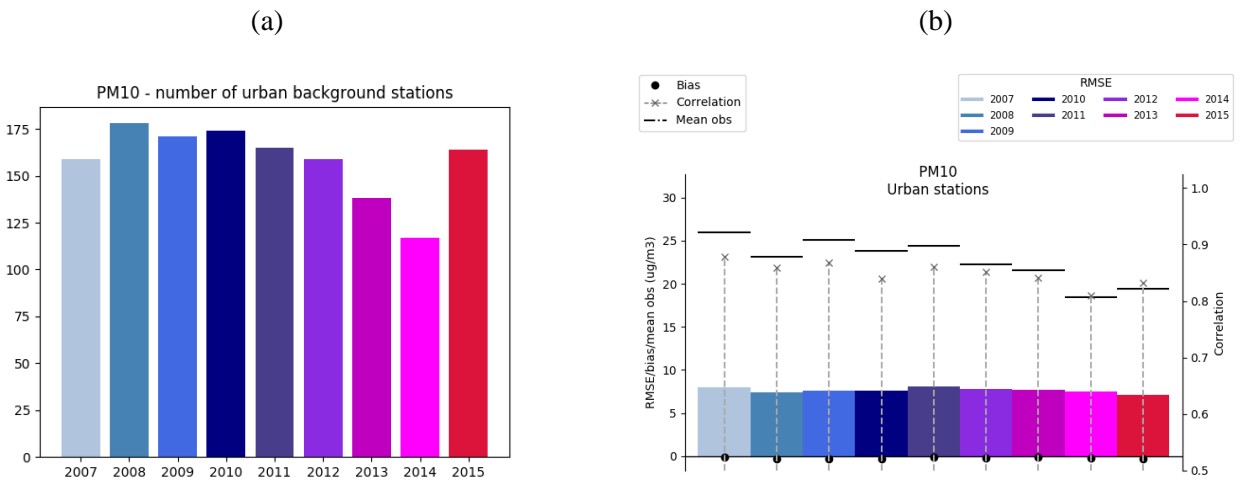

(a)                                                     (b)

Figure 2: PM₁₀: statistical indicators calculated using cross-validation technique on daily mean PM₁₀ values measured and estimated over URBAN background stations for the years 2007 to 2015. (a) number of rural stations for each year. (b) Bias (black circles), RMSE (coloured rectangles), correlation (grey crosses and the associated dashed lines) and mean observation (horizontal lines)

The scores show a good representation of the observations by the kriged data with correlations between 0.77 and 0.86 and RMSE of about 7 µg.m$^{-3}$, i.e between 30 % and 50 % of the annual mean PM10 concentration. The mean biases are particularly low for urban stations with values below -1 %. For rural stations the average bias is less than +3 µg.m$^{-3}$, i.e less than +15 %. The proportion between rural and urban stations varies between 1/3 and 1/10. The larger number of urban stations allows a better capture of the spatial variability of concentrations in urban environments.

Looking at the evolution of the scores over the years for rural stations, the number of stations available first increases from 2009 to 2012 before decreasing until 2014. In 2015 a new increase in the number of stations in France begins. For urban stations, the decrease starts earlier (2010) but the evolution is the same. The temporal evolution of the scores generally follows the number of stations with higher correlations and smaller relative mean biases and RMSE when more stations are available. Indeed, the greater the number of stations, the more representative the kriging technique will be of the real spatial variability. There are exceptions, however, as in 2015 for rural stations, with the second worst scores even though that year has the largest number of stations.

### 3.2. PM$_{2.5}$

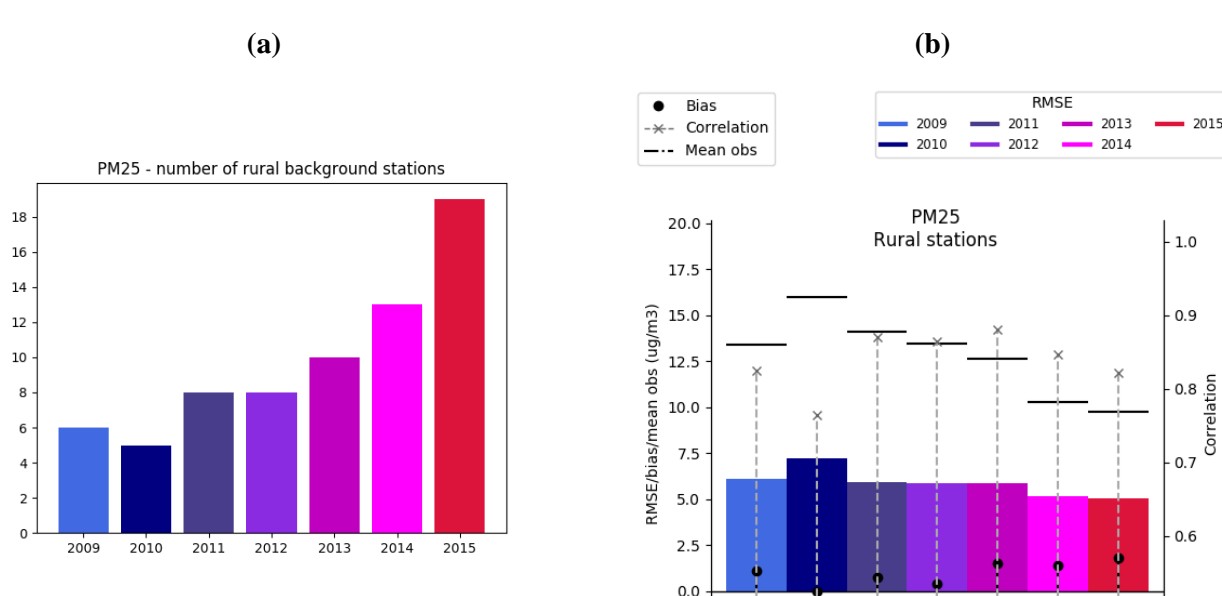

**Figure 3: PM$_{2.5}$: statistical indicators calculated using cross-validation technique on daily mean PM$_{2.5}$ values measured and estimated over RURAL background stations for the years 2009 to 2015. (a) number of rural stations for each year. (b) Bias (black circles), RMSE (coloured rectangles), correlation (grey crosses and the associated dashed lines) and mean observation (horizontal lines)**

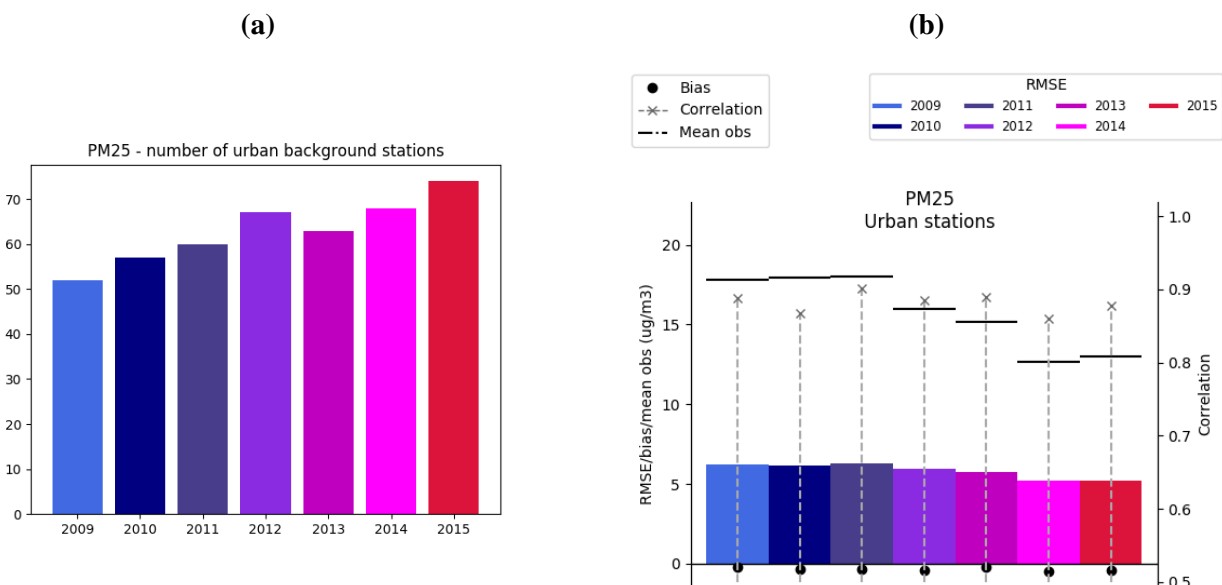

**(a)**        **(b)**

**Figure 4: PM$_{2.5}$: statistical indicators calculated using cross-validation technique on daily mean PM$_{2.5}$ values measured and estimated over URBAN background stations for the years 2009 to 2015. (a) number of rural stations for each year. (b) Bias (black circles), RMSE (coloured rectangles), correlation (grey crosses and the associated dashed lines) and mean observation (dotted horizontal lines)**

There are between half and a third fewer PM$_{2.5}$ stations than PM$_{10}$ stations. However, by using a co-kriging technique, the PM$_{2.5}$ mapping also benefits from PM$_{10}$ information, so that the correlations, mean bias and RMSE are almost similar to the PM$_{10}$ scores. The mean biases for rural stations do not exceed 20 % of the mean concentrations and are very low for urban stations (between 0 and -3 %). As for PM$_{10}$, this bias is systematically positive on rural stations (overestimation) and slightly negative over urban stations (underestimation). This is mainly related to the resolution of the data which smoothes the concentration gradients, giving a unique value on each grid (about 4 km horizontal resolution). For urban station, located close to PM$_{2.5}$ precursor emissions and generally having high concentration values, this smoothing effect leads to an underestimation. For rural areas far from emission precursors, the opposite is observed.

The correlation is generally higher than 0.8 and the RMSE does not exceed 7 µg.m$^{-3}$ (at maxima 50 % of the annual mean concentration).

### 3.3.  O₃

**(a)**

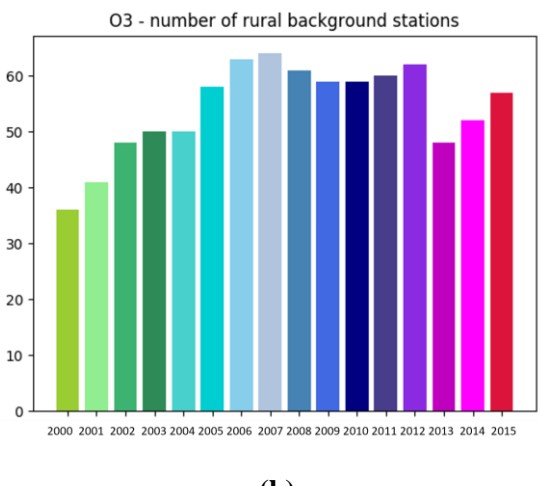

**(b)**

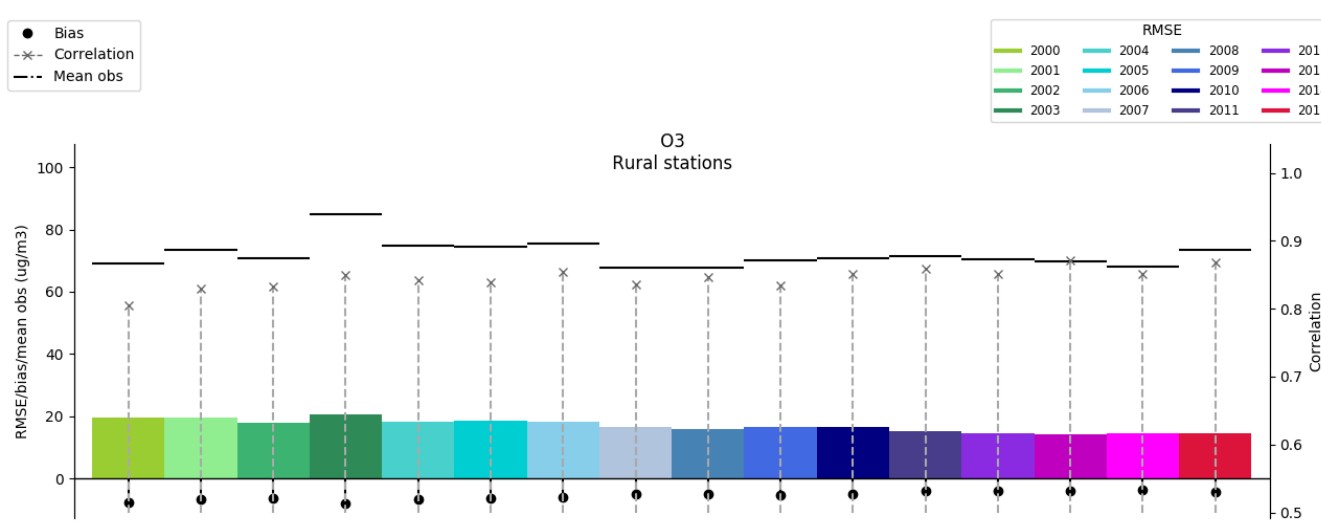

**Figure 5: O₃: statistical indicators calculated using cross-validation technique on daily mean O₃ values measured and estimated over RURAL background stations for the years 2000 to 2015. (a) number of rural stations for each year. (b)  Bias (black circles), RMSE (coloured rectangles), correlation (grey crosses and the associated dashed lines) and mean observation (horizontal lines)**

Comparison between estimated and observed ozone at rural stations shows good correlations (0.8 to 0.87), small relative mean negative biases (-4 to -8 %) and low RMSE (around 20 % of the annual mean concentration). Between 2000 and 2007, the number of rural stations increased, resulting in improved modelled concentration maps. The small decrease in the number of stations after 2007 do not penalise the scores for these years.

(a)

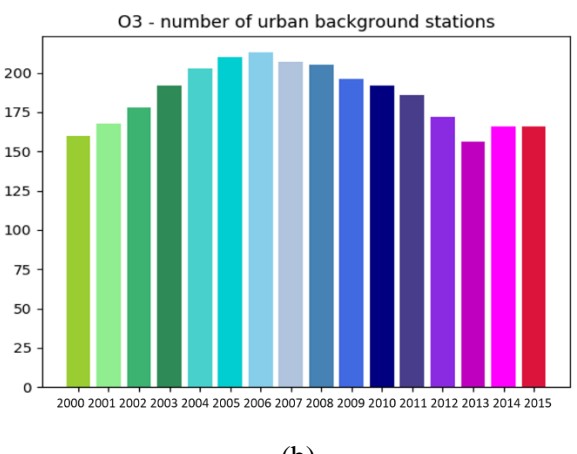

(b)

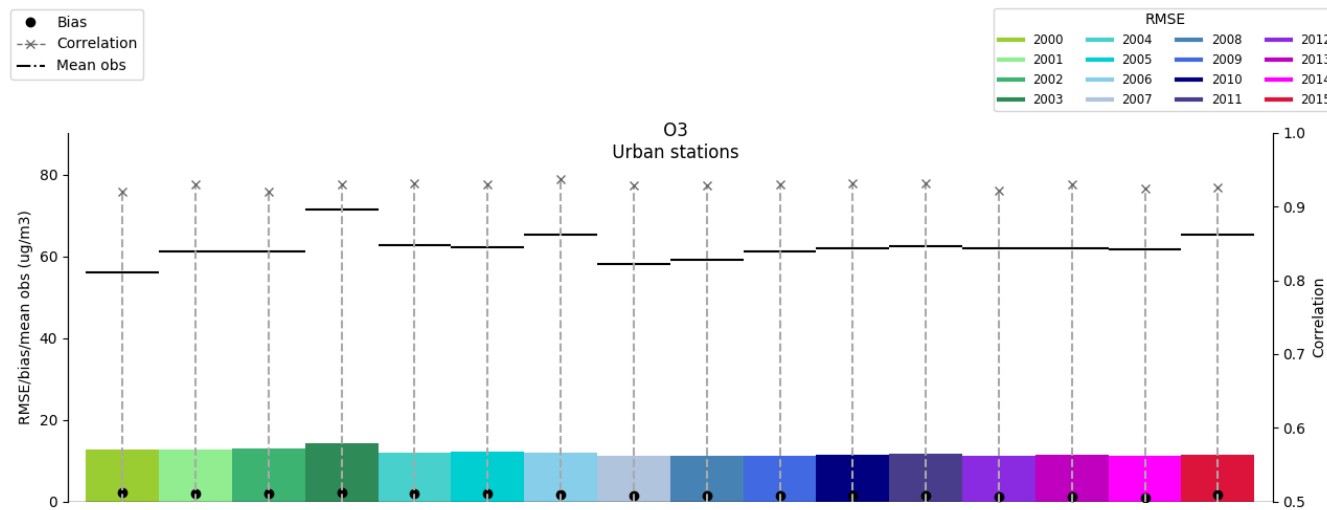

**Figure 6: O₃: statistical indicators calculated using cross-validation technique on daily mean O₃ values measured and estimated over URBAN background stations for the years 2000 to 2015. (a) number of urban stations for each year. (b) Bias (black circles), RMSE (coloured rectangles), correlation (grey crosses and the associated dashed lines) and mean observation (horizontal lines)**

The same conclusions can be drawn for the urban ozone scores. The higher number of urban stations even leads to slightly

10    better scores, with correlations above 0.9 for all years and relative mean positive biases not exceeding 5 %. A satisfactory

RMSE is also obtained for all years with values around 20 % of the annual mean concentration. It can be seen that the positive and negative biases are reverse with respect to the PM scores. Indeed, the highest $O_3$ values are generally observed in rural areas, where precursors have had time to produce $O_3$ and where $O_3$ destruction is lower than in urban areas. Therefore, the smoothing effect has the opposite effect to that of PM.

### 3.4. NO₂

**(a)**

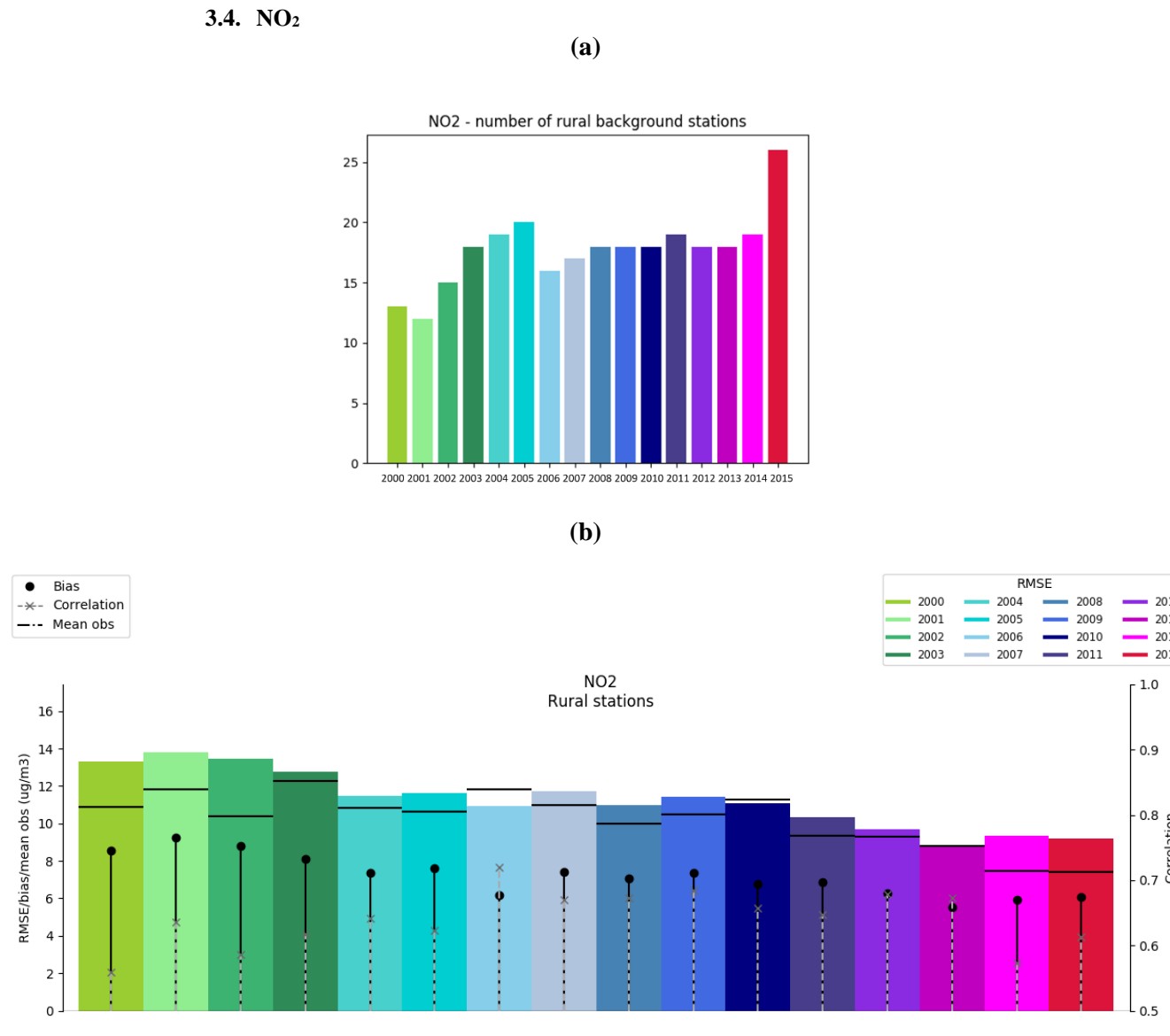

**(b)**

**Figure 7: NO₂: statistical indicators calculated using cross-validation technique on daily mean NO₂ values measured and estimated over RURAL background stations for the years 2000 to 2015. (a) number of rural stations for each year. (b) Bias (black circles), RMSE (coloured rectangles), correlation (grey crosses and the associated dashed lines) and mean observation (horizontal lines)**

Rural scores for NO$_2$ are worse for particles or O$_3$. The correlations are between 0.55 and 0.7 but above all, strong positive biases are observed for all years with an overestimation of the observations of 60 to 80%. This also affects RMSE scores that can exceed 100 % of the annual mean concentration. This poor performance can be explained by the strong spatial gradients in NO$_2$ concentrations due to its shorter atmospheric lifetime than O$_3$ or particles. There are too few rural stations to properly capture this variability in the kriging technique used here, so the urban stations have too much weight, and the raw model concentrations also overestimate rural concentrations.

**(a)**

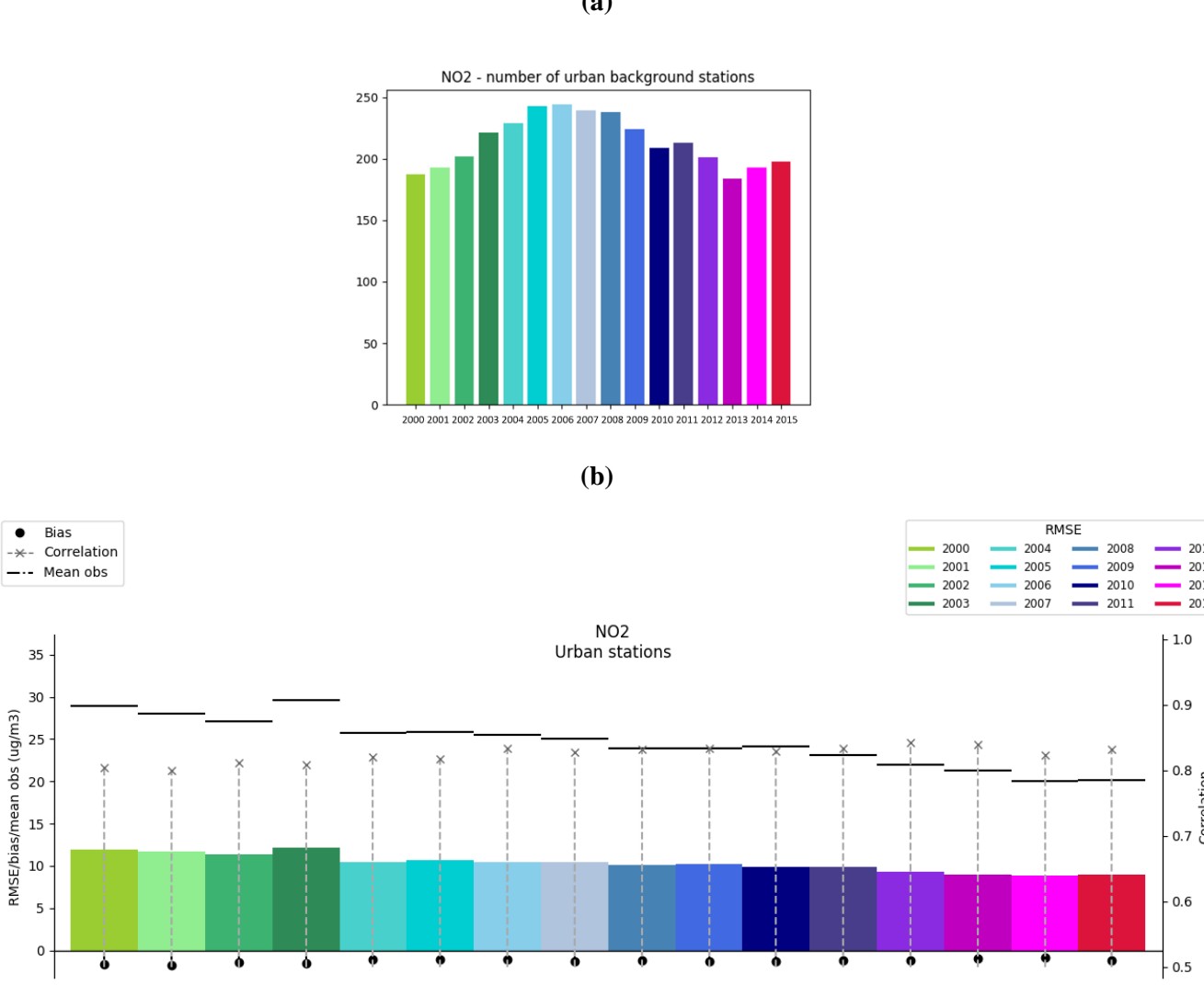

**(b)**

**Figure 8: NO$_2$: statistical indicators calculated using cross-validation technique on daily mean NO$_2$ values measured and estimated over URBAN background stations for the years 2000 to 2015. (a) number of urban stations for each year. (b) Bias (black circles), RMSE (coloured rectangles), correlation (grey crosses and the associated dashed lines) and mean observation (horizontal lines)**

The urban scores for $NO_2$ are much better than the rural scores. The correlations are around 0.8, the biases do not exceed -3.5 % and the RMSE is between 10 to 12 µg.m$^{-3}$ (less than 25 % of the annual mean concentration). The high number of urban background stations seems satisfactory to allow the kriging technique to correctly reproduce the spatial variability of $NO_2$ in urban background environments. It should be noted however that traffic stations are not used in the present analysis (neither as observational data to be compared with or included in kriging).

### 3.5. Comparison with other scores

In order to evaluate the added value of the kriging technique compared to the raw CHIMERE model simulations, the cross-validation scores can be compared to the raw model scores. **Table 3** shows the scores averaged over all years and all observations, without distinction of typology.

Table 3: Validation scores for the raw data and the kriged concentrations (cross-validation). Annual scores (bias, RMSE and the Pearson correlation coefficient r$^2$) are calculated over France for all year and all stations and are averaged.

|  | $NO_2$ | $O_3$ | $PM_{10}$ | $PM_{2.5}$ |
|---|---|---|---|---|
| RAW | | | | |
| Bias | -3.51 | 3.46 | -8.91 | -4.02 |
| RMSE | 12.97 | 17.26 | 12.63 | 8.73 |
| $R^2$ | 0.55 | 0.73 | 0.71 | 0.75 |
| KRIGED CONCENTRATION | | | | |
| Bias | -0.51 | -0.07 | -0.04 | -0.15 |
| RMSE | 10.41 | 12.54 | 7.64 | 5.83 |
| $R^2$ | 0.81 | 0.92 | 0.85 | 0.87 |

All scores are strongly improved by the kriging method of observations with CHIMERE in external drift. However, as can be seen in the previous figures, this improvement is more pronounced in urban areas than in rural areas, due to the much larger number of stations in urban areas, which makes the kriging more representative of these areas.

The cross-validation scores can also be compared with those obtained in Europe with other mapping methods. Chein et al. (2019) compared 16 algorithms to develop Europe-wide spatial models of $PM_{2.5}$ and $NO_2$, included linear stepwise regression, regularization techniques and machine learning methods. Those models were developed based on the 2010 routine monitoring data from the AIRBASE dataset, satellite observations, dispersion model estimates and land use variables as predictors. De Hoogh et al. (2018) also performed cross validation of their fine spatial scale land use regression models (also based on AIRBASE dataset, satellite observations, dispersion model estimates and land use variables as predictors) used in Europe for the year 2010. Results from their cross-validation are compared to our own cross-validation results in **Table 4**.

**Table 4: Validation scores for De Hoogh et al. (2018), Chein et al. (2019) and this study (Real et al. (2022)). The following scores are calculated by cross validation for the 3 studies : Pearson correlation coefficient $R^2$, the bias, and the Root Mean Square Error (RMSE).**

| | | De Hoogh et al., 2018 | Chein et al., 2019 | Real et al, 2022 |
|---|---|---|---|---|
| | $R^2$ | 0.57 | 0.57 - 0.62 | 0.81 |
| $NO_2$ | RMSE | 9.51 | 9 - 9.5 | 10.41 |
| | Bias | | | -0.51 |
| | $R^2$ | 0.58 - 0.68 | 0.48 - 0.63 | 0.87 |
| $PM_{2.5}$ | RMSE | 2.97 - 3.3 | 3.1 - 3.9 | 5.83 |
| | Bias | | | -0.15 |
| | $R^2$ | 0.63 | | 0.92 |
| $O_3$ | RMSE | 6.87 | | 12.54 |
| | Bias | | | -0.07 |

The comparison of performance in these three studies is of course limited by the fact that the spatial coverage differs: in De Hoogh et al. (2018) and Chein et al. (2019), the cross validation is computed over the whole of Europe. In this study, the performances are assessed over France.

For all pollutants the spatial correlation (R2) is better in our study. In the same time, higher RMSE are also found for our study. This may be due to a larger bias, but we also demonstrated in our paper that the bias was very small, except at rural NO2

stations. Since the RMSE score also depends on the absolute concentrations, the different spatial coverage may also play a role. The lower RMSE over Europe could be an artefact of including areas where absolute concentrations of NO2, PM2.5 or O3 are lower than over France.

The validation scores obtained, as well as the comparison with raw data and with other mapping method, allow us to be confident about the validity of the concentrations obtained and their good representativeness of background concentrations, in

particular in urban areas. A point of vigilance appears however when it comes to the representativeness of rural NO2 concentrations which are overestimated in our results.

## 4. Results and discussion

After ensuring the validation of the kriged concentration data, yearly indicators, trend over years and human exposition are calculated. Hourly concentrations fields are produced from 2000 to 2015 for $NO_2$, $O_3$ and $PM_{10}$, however, as explain in section

2, for $PM_{10}$ only annual mean indicators maps are produced before 2007. $PM_{2.5}$ hourly concentrations are calculated for year 2009 to 2015 due to the limited number of background stations available before 2009.

### 4.1 Concentration maps and trends

All the indicators presented in section 2 are calculated but the following section focus on the annual averaged concentrations of $PM_{10}$, $PM_{2.5}$, $NO_2$ and $O_3$, as well as SOMO35 and AOT (two indicators associated with $O_3$), for which mapped data are presented. These indicators are presented in this paper and available on a zenodo repository and on an online map library (see

5    section 5).34 Trend analyse over the period is performed by calculating the Sen-Theil regression slope for each grid point on the domain. To characterise the significance of these trend slopes, the 95 % confidence interval is calculated. This confidence interval represents the lower and upper values above or below which there is ( 95 %) confidence that the trends will occur. The smaller the confidence interval, the more statistically significant the trend. Large confidence intervals are considered as unrepresentative, especially those containing 0. Trend slopes and confidence intervals are calculated for each grid point in the

10   domain and country averaged values are also given in Table 5.

**Table 5: country averaged slope and its 95 % confidence interval**

| Indicator | Mean tendency slope (or mean trend) in $\mu g.m^{-3}.year^{-1}$ | Mean 95 % confidence interval (in $\mu g.m^{-3}.year^{-1}$) |
|---|---|---|
| $PM_{10}$ - avg annual | -0.8 | [-0.5 ; -1.09] |
| $PM_{2.5}$ - avg annual | -0.87 | [-0.48 ; -1.41] |
| $O_3$ - avg annual | 0.32 | [0.005 ; 0.59] |
| $O_3$ - SOMO35 | -5.52 | [ -102.7 ; 76.7 ] |
| $O_3$ - AOT | -142 | [-641 ; 315] |
| $NO_2$ - avg annual | -0.32 | [-0.3 ; -0.63 ] |

### 4.1.1. PM₁₀

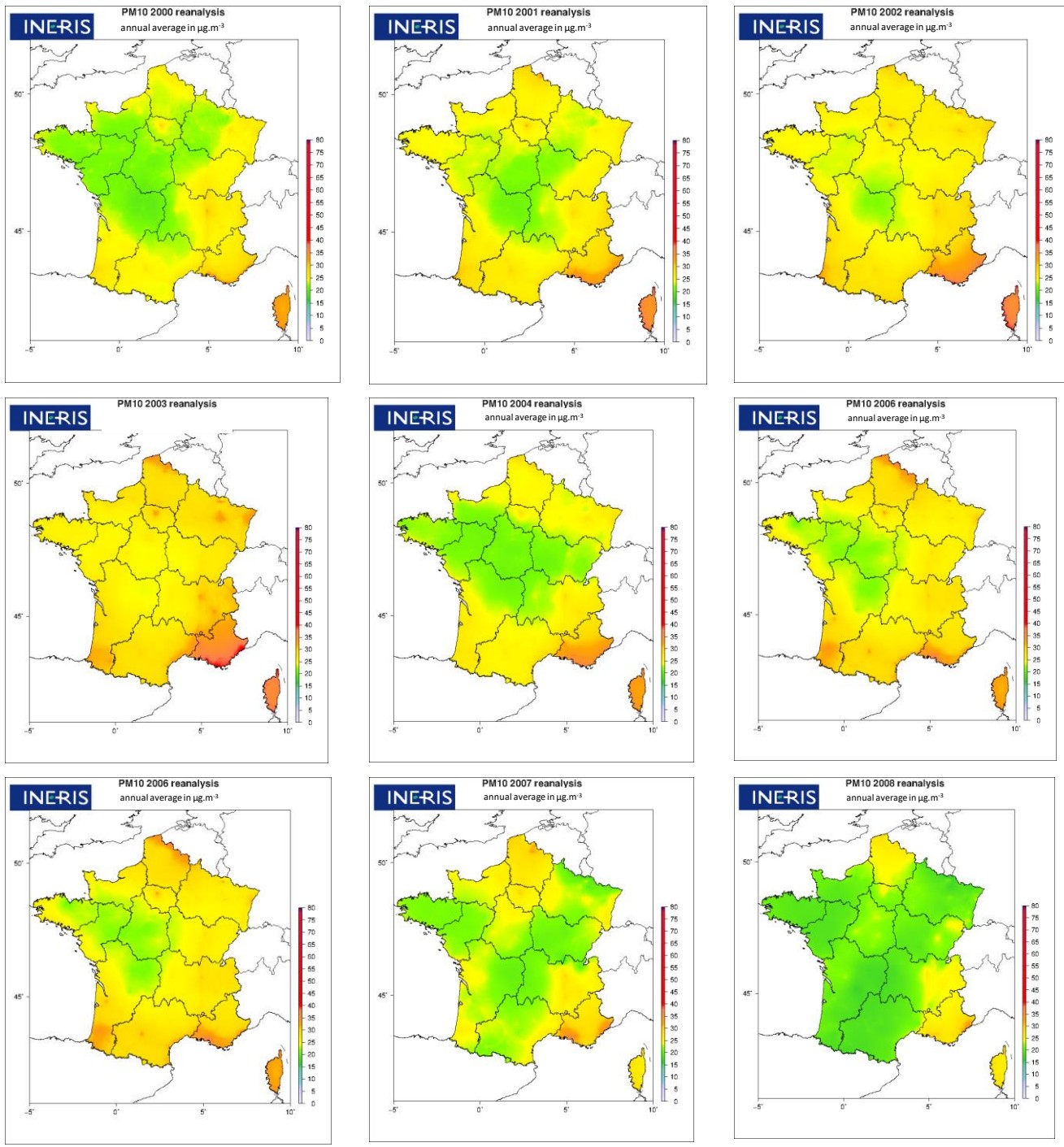

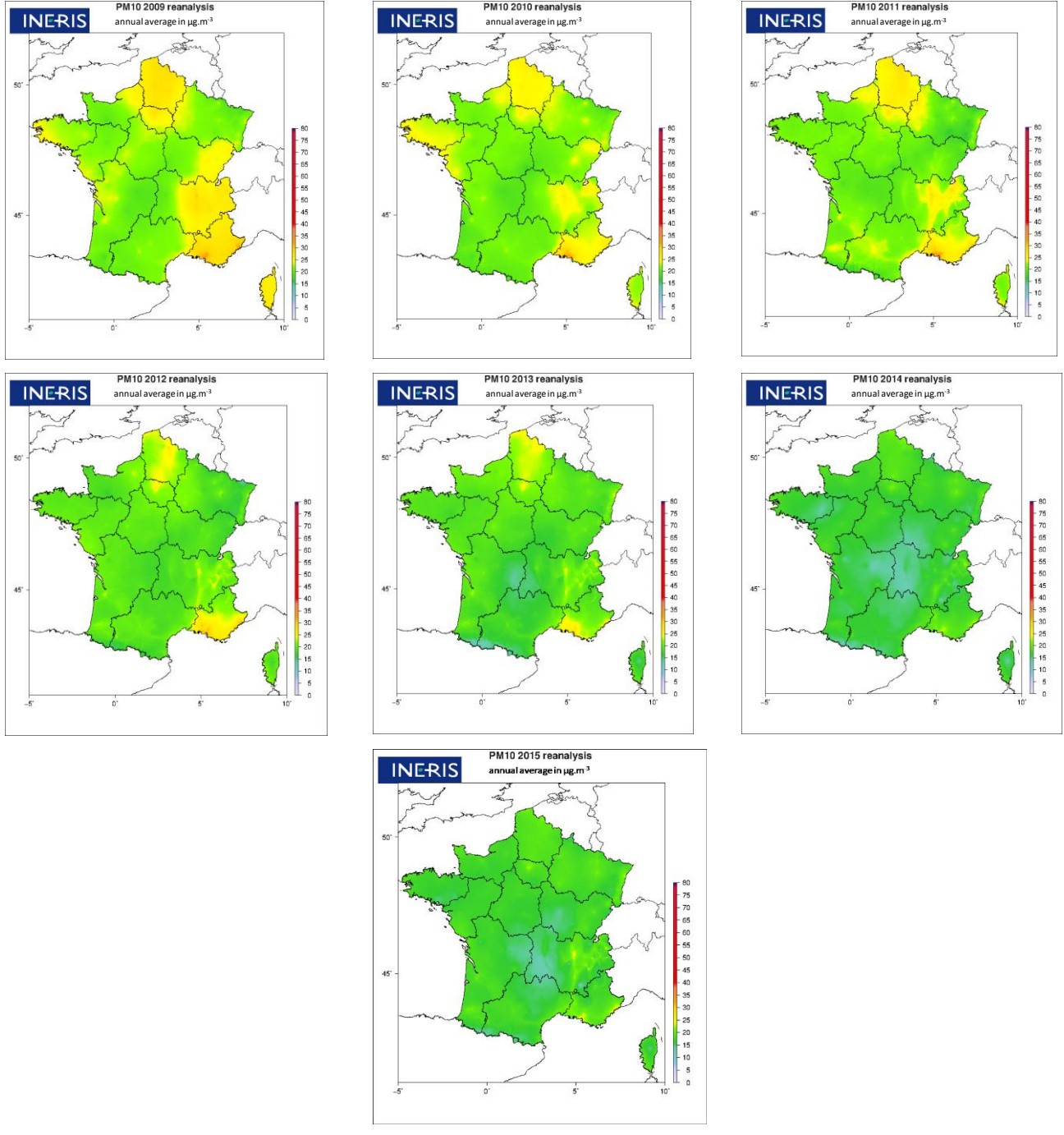

**Figure 9: PM₁₀ annual mean concentrations from 2000 to 2015. Concentrations are obtained by combination between regional modelling and observations**

Maps of annual average PM$_{10}$ concentration maps are presented in **Erreur ! Source du renvoi introuvable.**. for the period 2000-2015. The resolution of the grid (around 4km) allows to see patterns such as interconnected cities, especially in the latest years for which the patterns of large inter-regional concentrations are decreasing. The impact of meteorological conditions is also visible through the interannual variability. For example, the 2003 heatwave year is associated with higher PM$_{10}$ levels due to increased formation of secondary aerosols.

Figure 10 shows the mapped trends in annual average PM$_{10}$ expressed as Sen-Theil regression slope in µg.m$^{-3}$ per year and calculated over the period 2000-2015.

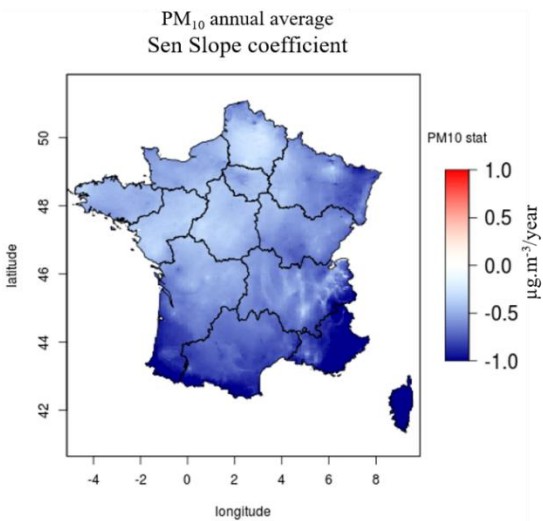

**Figure 10: Trends in PM$_{10}$ annual mean concentration. Sen slope coefficient (µg.m$^{-3}$/year) calculated over the period 2000-2015**

There is a downward trend in PM$_{10}$ annual mean concentrations everywhere in France, and in particular in the regions with the highest PM$_{10}$ concentrations at the beginning of the period: the South of France (East and West), the Auvergne-Rhône-Alpes region, the East (Grand-Est) and the extreme north of France. A country-averaged downward trends in PM$_{10}$ concentrations of -0.8 µg.m$^{-3}$ per year is estimated over the period 2000-2015 (spatial average of the trends calculated on each grid point). This trend is statistically significant on average over France with a narrow 95%-confidence interval ([-0.50;-1.09]) that does not include zero (see Table 5) and applies to almost all grid points (maps of confidence interval, not shown here) Taking the year 2000 as the base year, this amounts to a 39% reduction. In a study conducted for France over the period 2000-2010, Malherbe et al. (2017) estimated a downward trend that was twice as small (0.4). This reflects the accelerated decline in concentrations in France in recent years.

This significant downward trend is the result of the decrease in primary pollutant emissions over these 16 years in response to emission reduction measures. From 2000 to 2015, primary PM$_{10}$ emissions over France have been reduced by 39 %, as well as emission of PM$_{10}$ precursors such as NO$_x$ emissions (-56 %) and SO$_x$ emissions (-87 %) (data calculated by the CITEPA and extracted from the 2015 French national air quality report https://www.statistiques.developpement-durable.gouv.fr/sites/default/files/2018-10/datalab-bilan-de-la-qualite-de-l-air-en-france-en-2015-octobre-2016-c.pdf).

### 4.1.2.  PM2.5

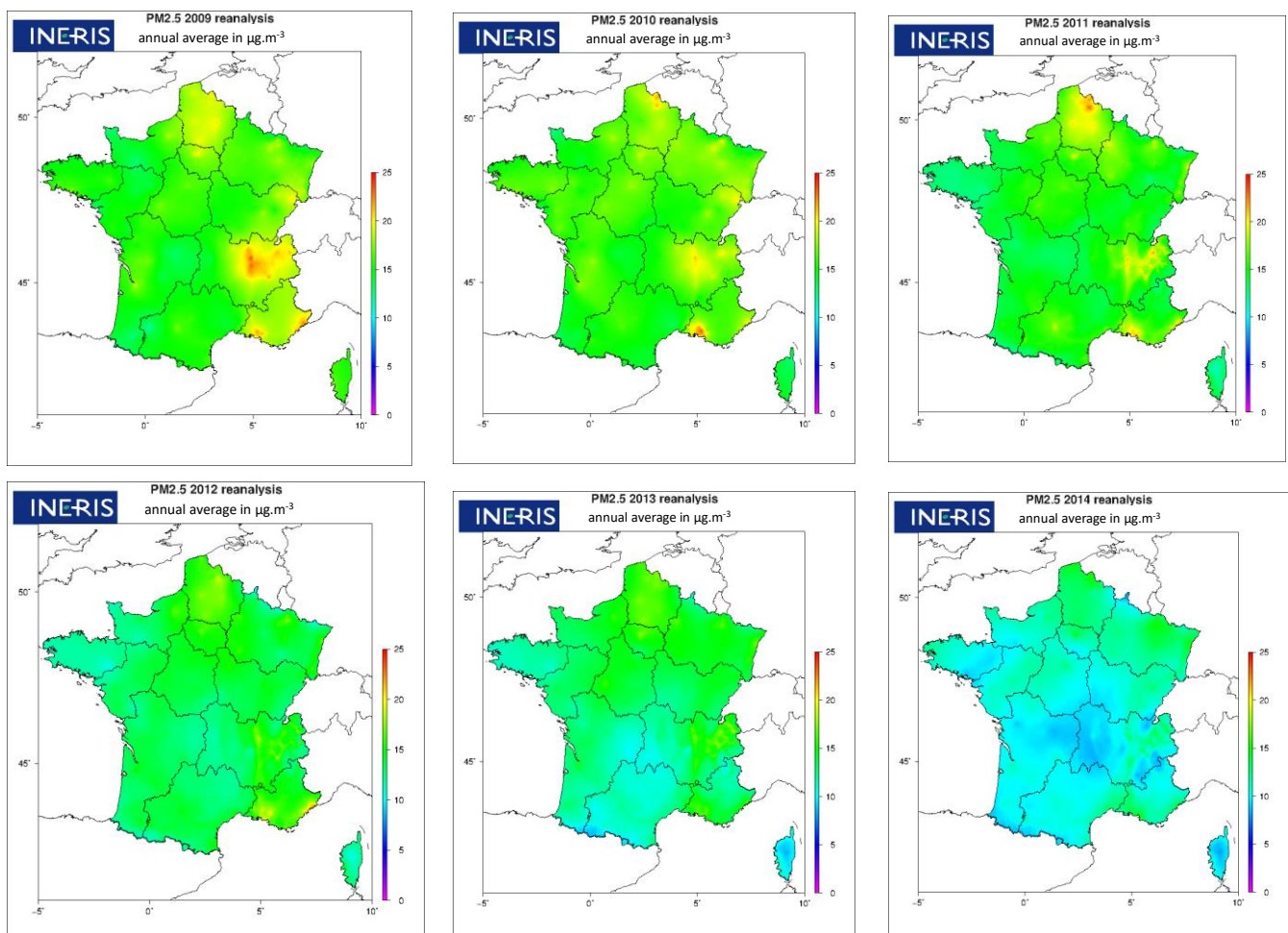

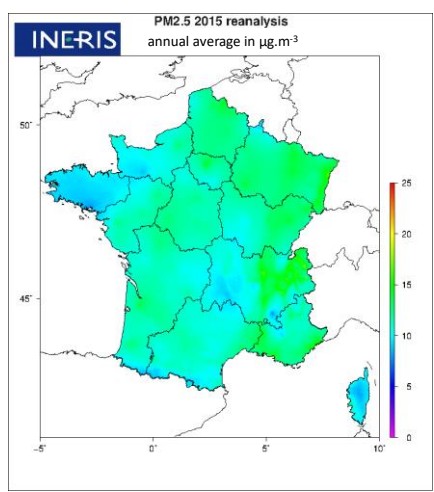

**Figure 11: PM₂.₅ annual mean concentrations from 2009 to 2015. Concentrations are obtained by combination (kriging) between regional modelling and observations.**

The highest PM$_{2.5}$ values are observed at the beginning of the period and are more concentrated in the main source regions than PM$_{10}$. Significant reductions in annual average background concentrations are observed over the years. The Sen slopes coefficients calculated for the annual average PM$_{2.5}$ (Figure 12.) over the period show negative trends over the whole territory, more pronounced over the South-East region, the Auvergne-Rhone-Alpes region, Northern France and Brittany. A downward trend of -0.87 µg.m$^{-3}$ per year on a national average is calculated, again with statistical significance (95 % interval of [-0.48;-1.41] which does not contain zero). Taking 2009 as a reference year, this amounts to a 35% decrease in 7 years. As for PM$_{10}$, this negative trend is associated with the reduction of primary PM$_{2.5}$ emissions and in PM$_{2.5}$ precursors emissions (SOx, NOx and VOC).

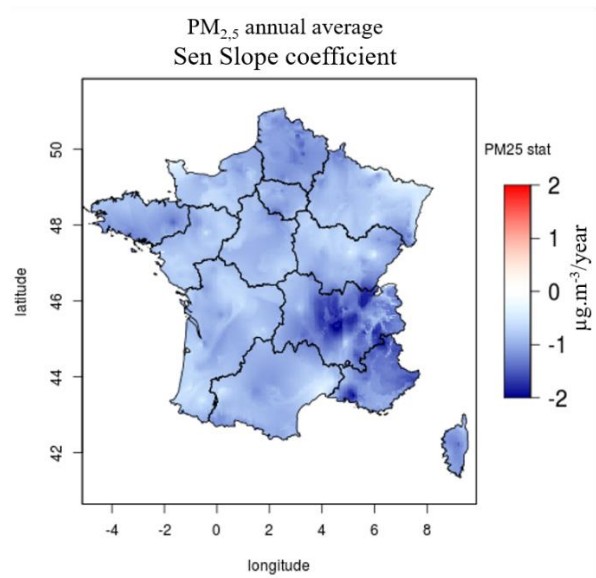

**Figure 12: trends in PM₂.₅ annual mean concentration. Sen slope coefficients (µg.m⁻³/year) calculated over the period 2009-2015.**

### 4.1.3. Ozone

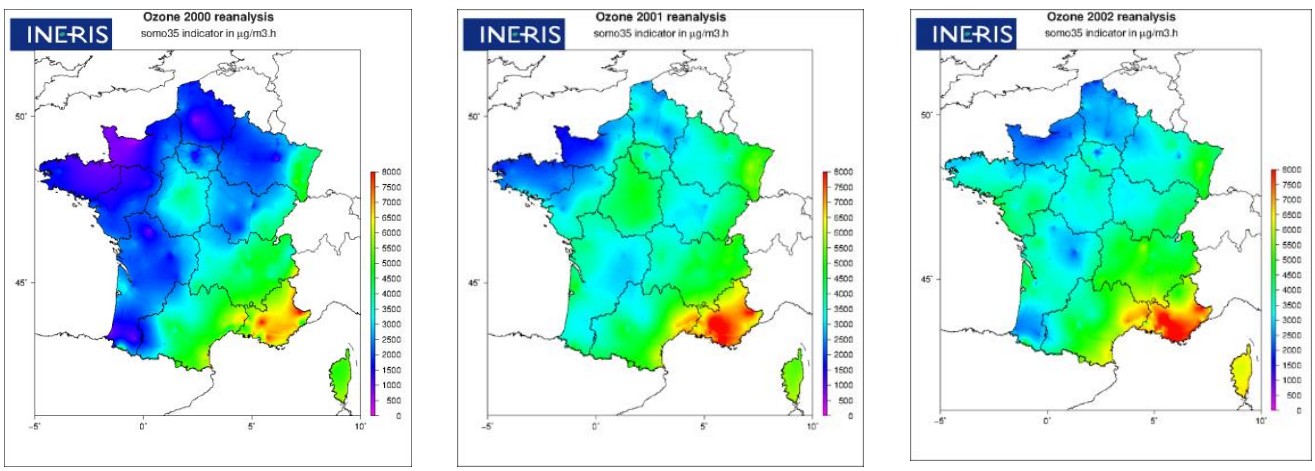

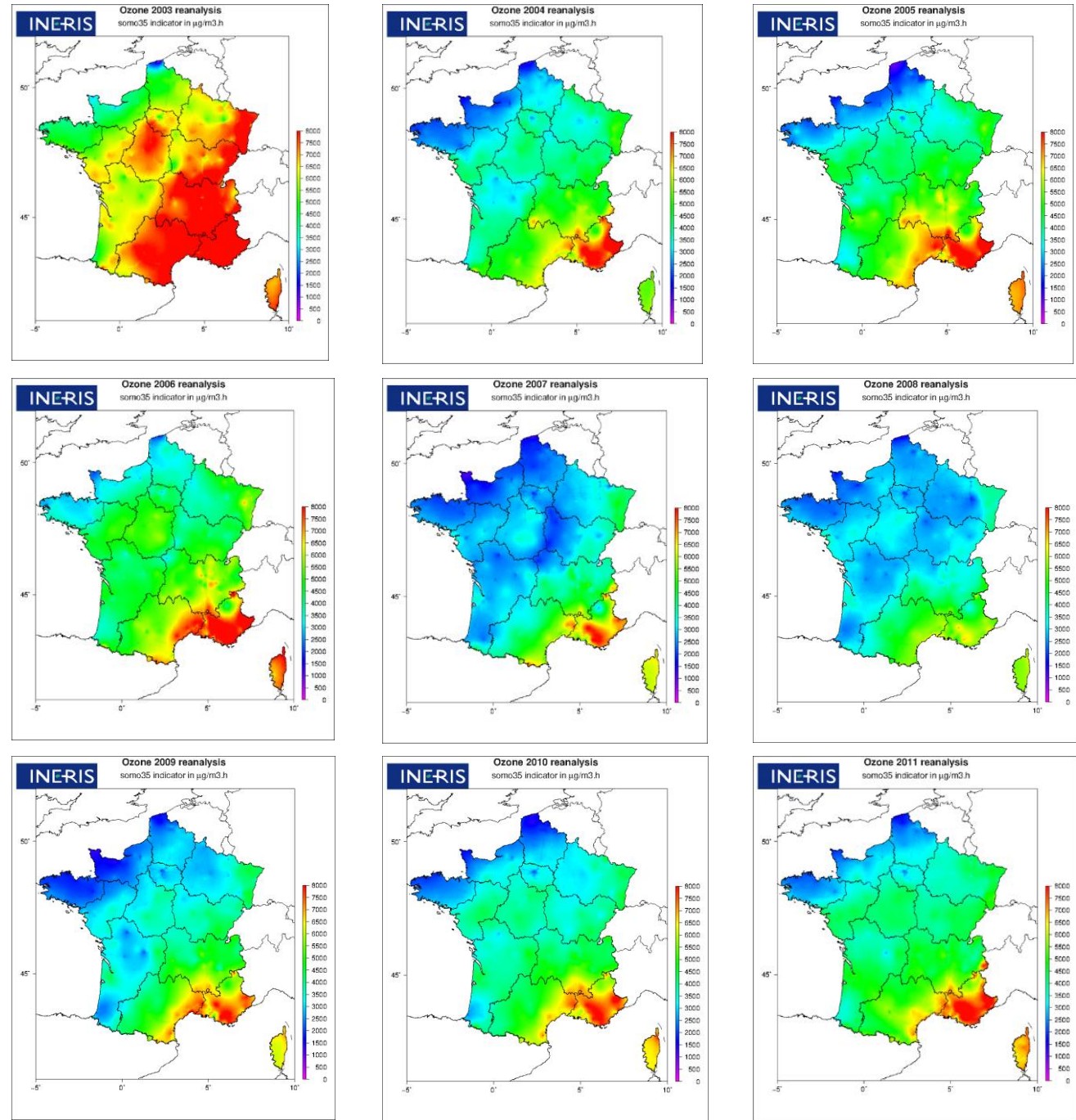

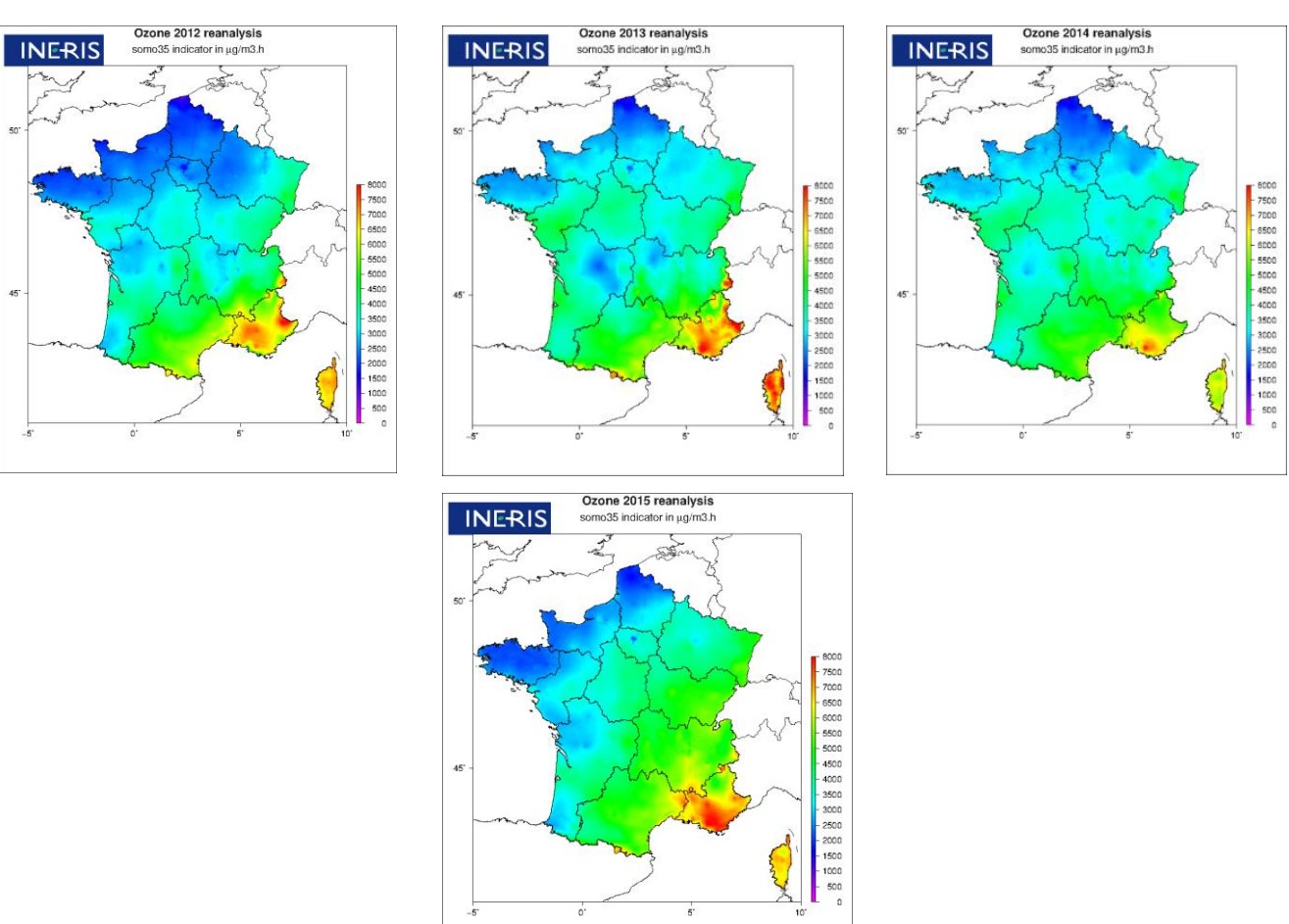

**Figure 13: SOMO35 indicator for the period 2000 to 2015. Ozone concentrations are obtained by combination (kriging) between regional modelling and observations.**

The SOMO35 indicator shows strong interannual variability. $O_3$ is a photochemical pollutant produced by secondary reactions in the presence of $NO_x$, VOC and sunlight. The hot year 2003 is distinguished by a very high SOMO35 over almost the entire territory. For each year, the highest SOMO35 is found in the south-eastern France and to a lesser extent in the Alsace region. The trends in SOMO35, annual average $O_3$ and AOT40 over the years are shown in

a)   Yearly mean concentrations                              b)   SOMO35

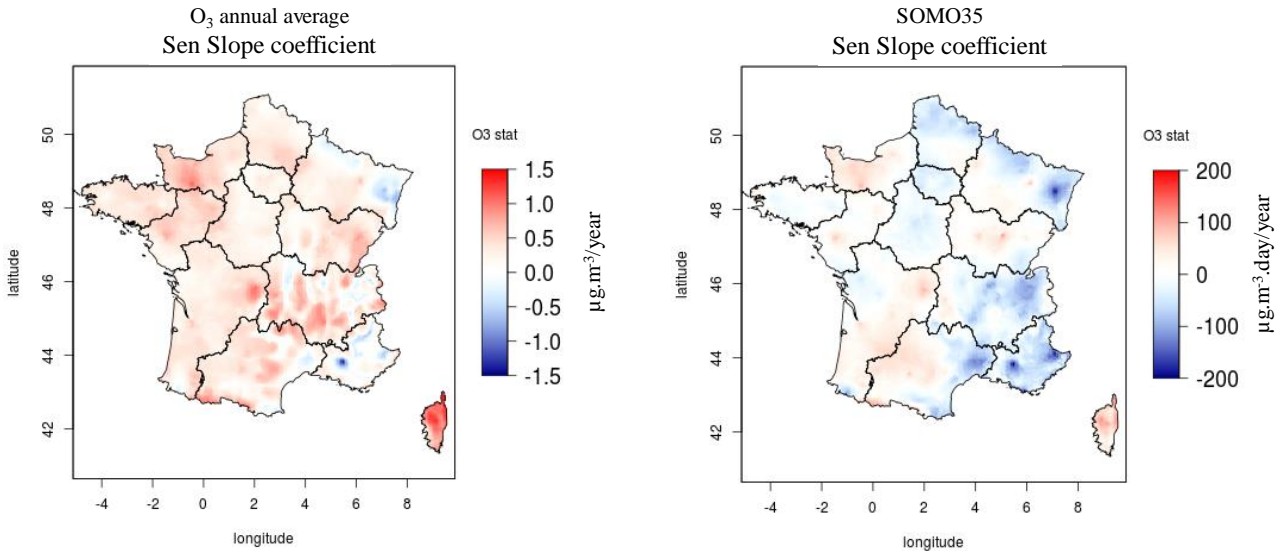

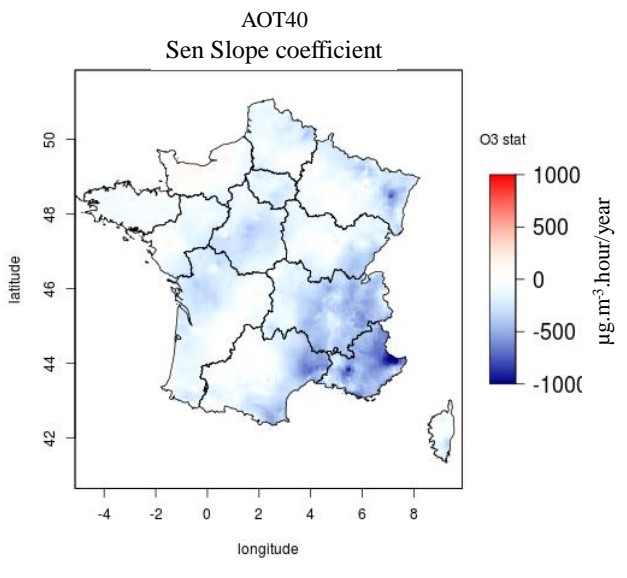

**Figure 14** for the period 2000-2015.

a)d)Yearly mean concentrations                          b)e)SOMO35

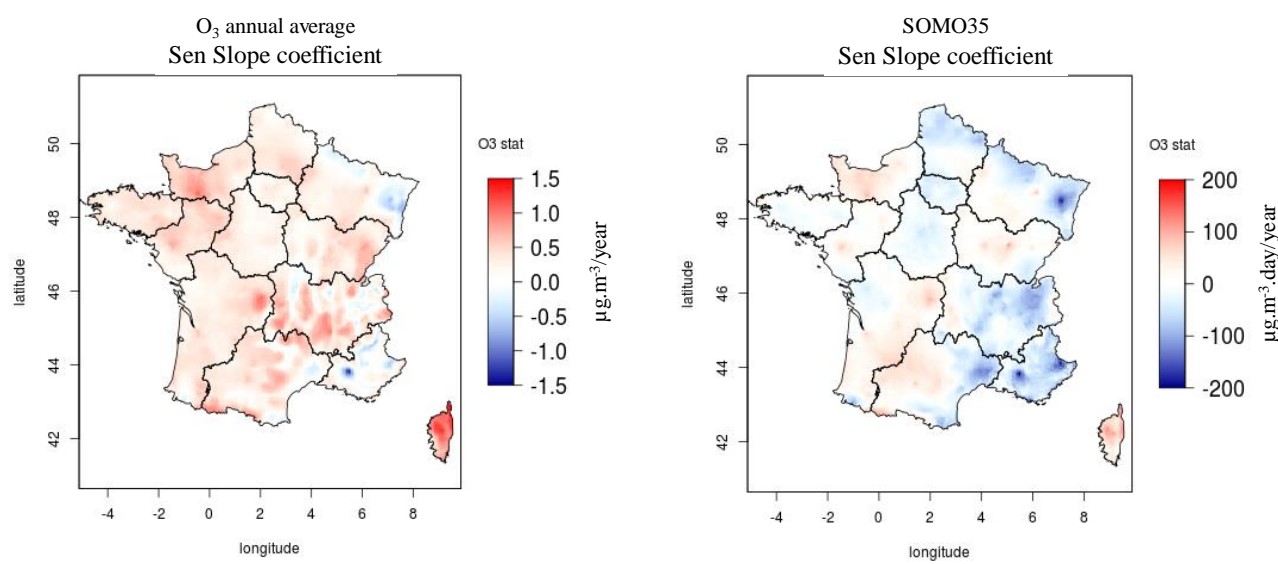

e)f) AOT40

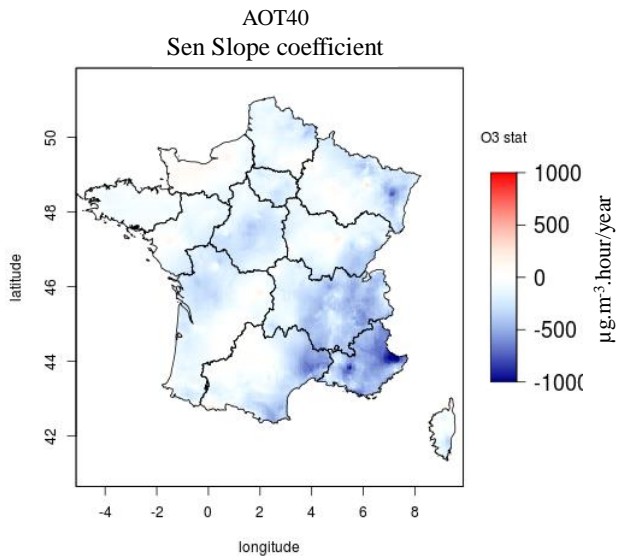

**Figure 14: Trends in annual mean O₃ concentrations in μg.m⁻³.year⁻¹ (a), SOMO35 in μg.m⁻³.day.year⁻¹ (b) and AOT40 in μg.m⁻³.hour.year⁻¹ (c) indicators. Sen slope are calculated over the period 2000-2015.**

For the O₃ average annual concentration, small positive trends are found over France. Two exceptions are the south-east (PACA region) and the Grand-Est region (East of France), i.e the regions with the highest O₃ concentrations, showing
5   negative trends. Averaging over France, this leads to a positive trend of 0.32 $\mu g.m^{-3}$. $year^{-1}$ which corresponds to an increase of 6.5% over 16 years. The same order of magnitude was found for the period 2000-2010 by Malherbe et al. (2016). Both negative (in South of France) and positive trends are significant according to the mapped 95 % confidence interval (not shown). SOMO35 and AOT40 indicators, which are indicators with a threshold value below which concentrations are not taken into account, show mostly negative trends. However according to the value of the mapped 95 % confidence interval
10   (not shown here) on most grid points, the confidence interval is wide and contains zero, indicating a lack of significance of the calculated trends. These results are consistent with other European studies (EMEP 2016, Malherbe et al., 2017) that show an increase in background concentrations and a decrease in O₃ peaks.

4.1.4.        **NO₂**

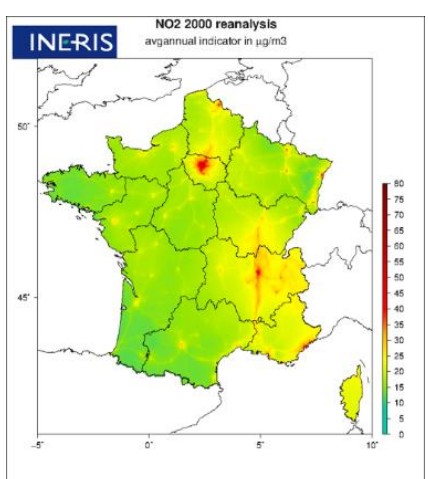 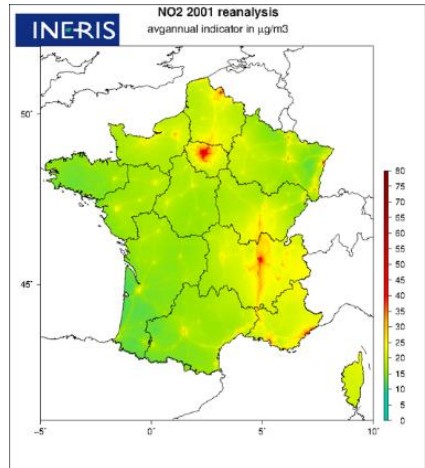 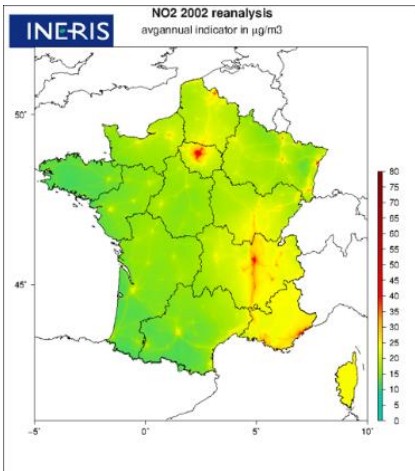

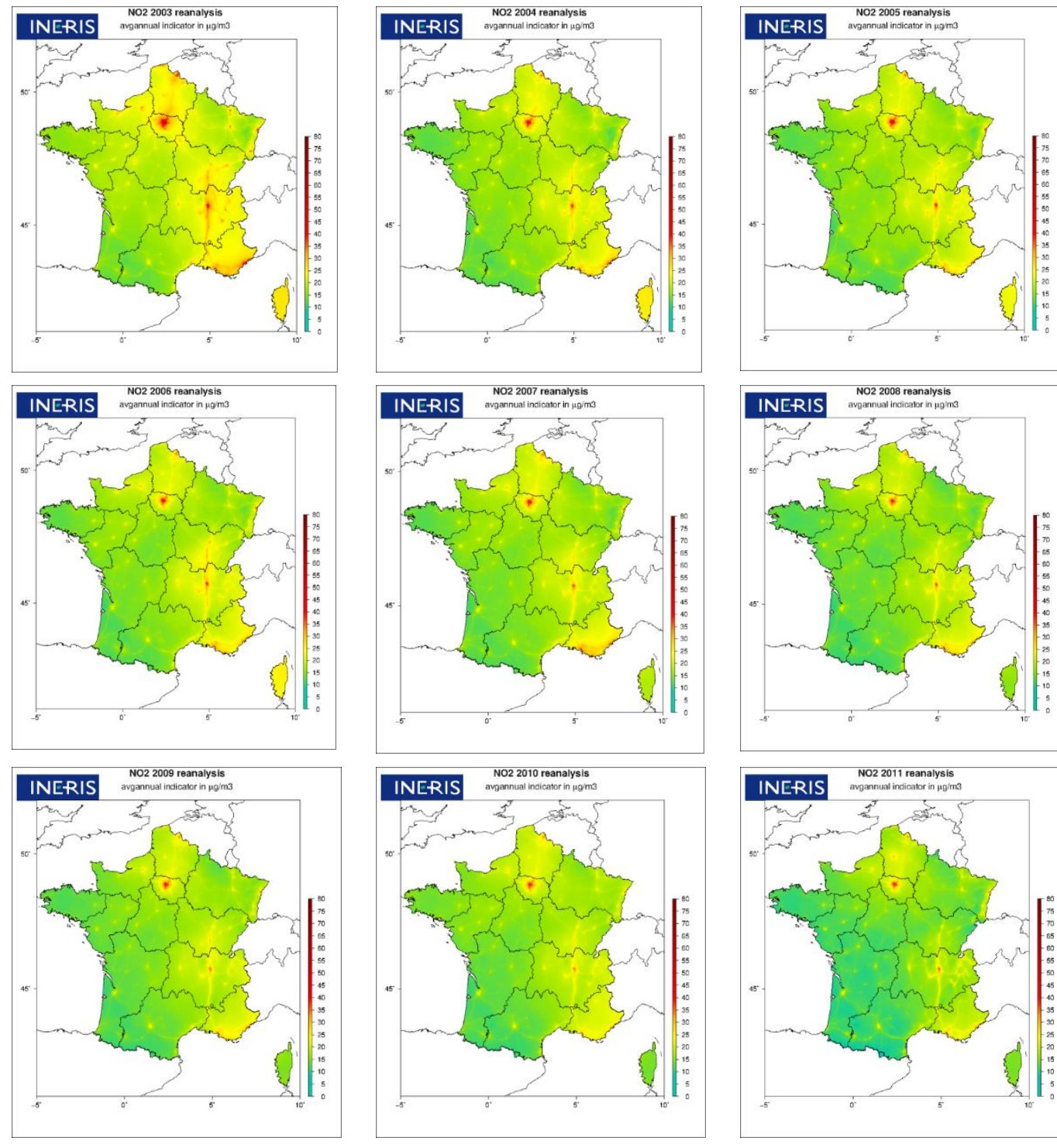

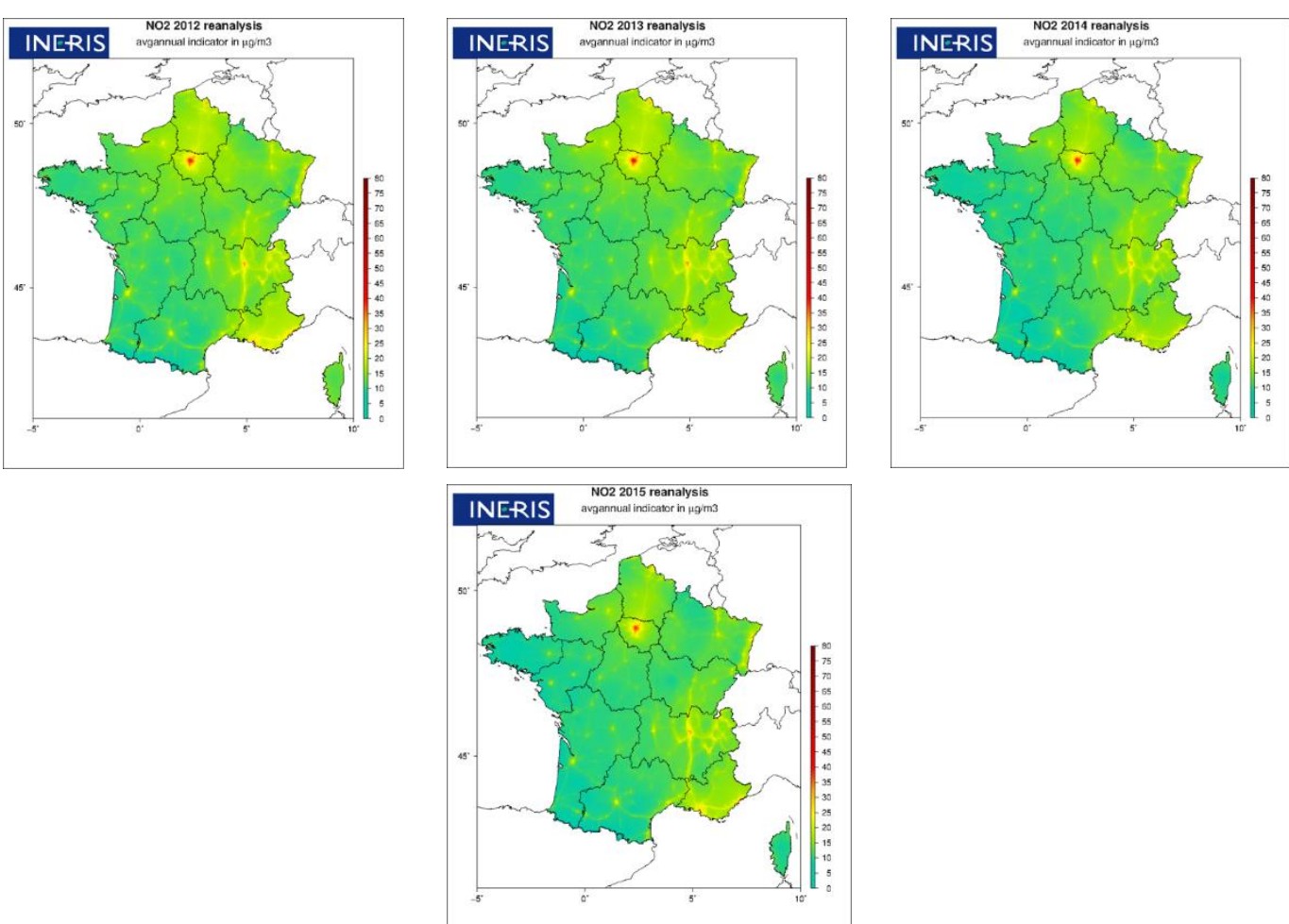

**Figure 15: NO₂ annual mean concentrations for the period 2000 to 2015. NO₂ concentrations are obtained by combination between regional modelling and observations.**

NO₂ is mainly emitted by road transport. All maps show the same pattern, with cities and interconnected major roads showing the highest NO₂ concentrations. Trends over the period 2000-2015 are shown in Figure 15. Decreases in NO₂ concentrations are observed in both rural and urban areas throughout the country. However, we recall that rural levels were found to be overestimated with our approach (see 3.4). The decrease is more important when NO₂ concentrations are high. As with PM₂.₅, these results highlight the combined benefit of large-scale emission management policies that target emission sectors and locally-oriented policies.

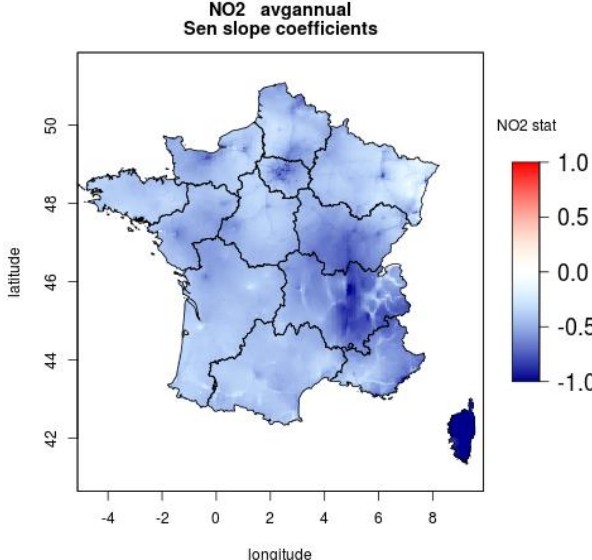

**Figure 16: Trends in yearly mean NO₂ concentrations. Sen slope coefficients (µg.m⁻³/year) are calculated over the period 2000-2015.**

On average, a significant negative trend of -0.46 µg.m⁻³ is calculated over France, with a narrow 95 % confidence interval (see Table 3). This downward trend is slightly stronger than that calculated in Malherbe et al. (2017) over the period 2000-2010 over France (-0.37 µg.m⁻³·year⁻¹) and corresponds to a reduction of about 30% (taking 2020 as the base year).

## 4.2 Exposure trends

Population-weighted annual average concentrations are good estimates of population exposure as they give more weight to the air pollution where people mainly lived. Here, the country-averaged population weighted concentrations of NO₂, PM₂.₅ and SOMO35, which are the 3 main indicators used to calculate health impact, are calculated for each year, from the hourly kriged mapped data over France. For one pollutant, it is obtained adding the result of multiplying the concentration by the population on all the country's grids, then dividing by the total population of the country. The population database used in this study is the LCSQA national population database (Létinois et al., 2014) established for the year 2015. It is based on detailed files from the French Ministry of Finance with information at building level. It is important to note that the French population used here has not varied over the years. The French population increased by about 10 % between 2000 and 2015. However, if we considered that the demographic evolution is homogeneous over the country (the urban/rural ratio has only increased by about 2.5% in France over the same period), the weighted population concentration on national average should be the same whatever the year of the population database.

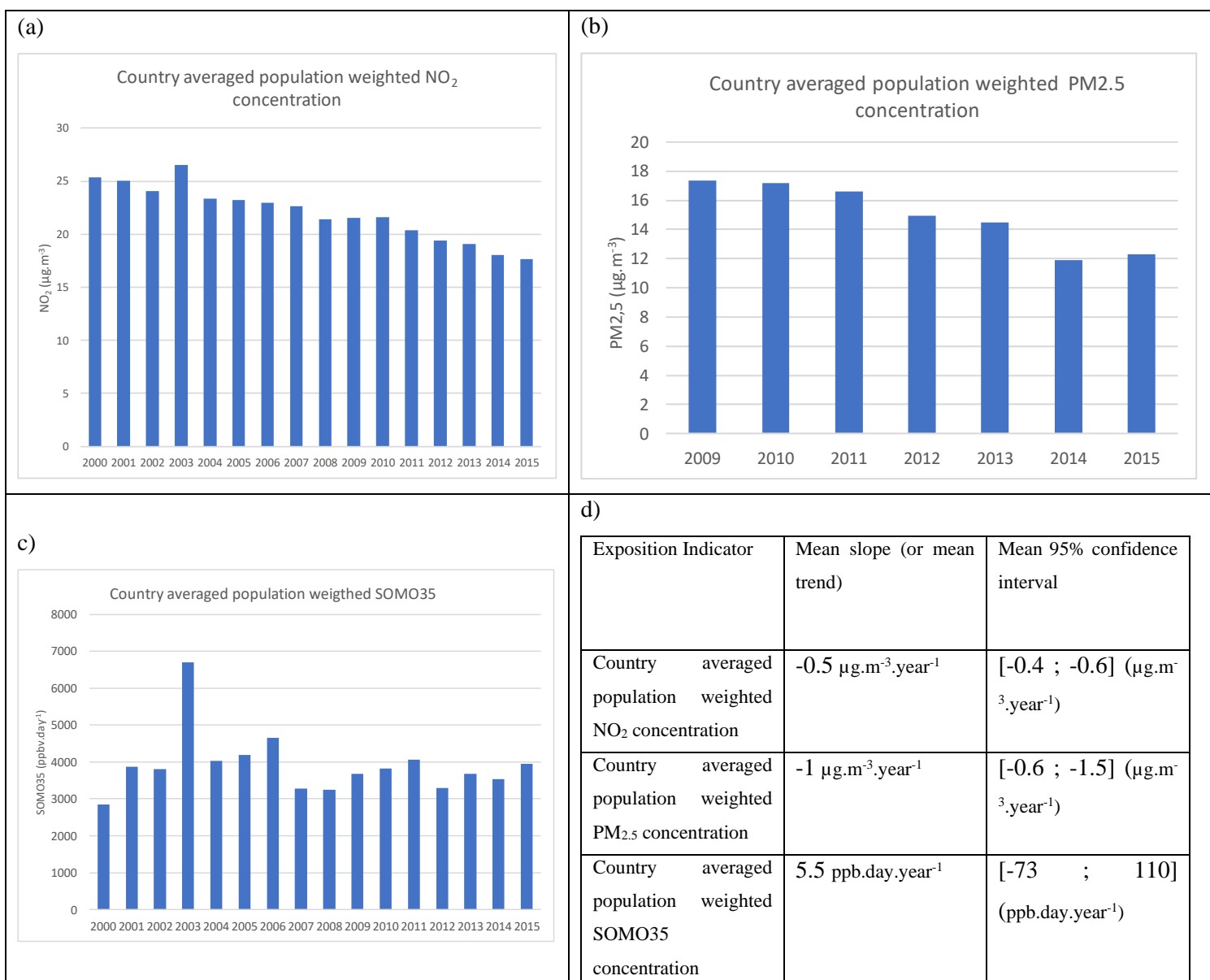

**Figure 17: Yearly evolution of the country averaged population weighted of (a) NO₂ concentration (b) PM₂.₅ concentration c) SOMO35. Trends and 95% confidence intervals are calculated (d).**

As for the concentrations, a very clear downward trend is observed for population-weighted $NO_2$ with a trend of -0.5 µg.m⁻³.year⁻¹ and a narrow 95 % confidence interval: ([-0.4,-0.6]), i.e a reduction of about 30 % in 16 years. A downward trend of -1 µg.m⁻³.year⁻¹ is also clearly calculated for $PM_{2.5}$ (95 %-confidence interval: [-0.6,-1.5]) over the period 2009-2015, i.e a reduction of about 31 % in 7 years. In contrast, there is no clear trend for the SOMO35 indicator over the period 2000-2015.

When the abovementioned indicators are multiplied by the total population (to obtain the total exposure, i.e the sum of the population weighted over a country), the outcome indicators are those used to calculate the health impact assessment based on dose-response functions, as suggested by the WHO review of "Health Risks of Air Pollution in Europe" (WHO 2013), described in Holland (2014 a and b). Exposure to SOMO35, anthropic $PM_{2.5}$ and $NO_2$ (with or without threshold depending on the health impact indicator) contribute to both morbidity and mortality impacts. For example in France, they were used in the PREPA-evaluation study for which about fifty political measures to be implemented in France were evaluated and ranked on different criteria, such as air quality impact, health impact and cost-benefit assessment (Schucht et al., 2018). At constant population evolution, the trends are similar between both indicators (total exposure and population weighted average concentration). However, the evolution in population (even if it is homogeneous over the territory) has an impact on the total exposure of the population. Therefore, we expected a reduced impact on health impact assessment compared to those on population weighted concentrations.

## 5. Data availability

Mapped regulatory indicators and exposure data for all 15 years and the 4 pollutants described here are available on a zenodo repository under the Netcdf format (version n°4) and csv format for data at the municipal or regional level. The DOI link for the dataset is http://doi.org/10.5281/zenodo.5043645 (Real et al., 2021). It is also available through a web-based map library (https://www.ineris.fr/fr/recherche-appui/risques-chroniques/mesure-prevision-qualite-air/20-ans-evolution-qualite-air). The web-based map library is intended to be updated annually. Those data have been provided to several research teams with different field of expertise ranging from epidemiology, to environmental economics and atmospheric science. Most of this work is still in progress, but others are the subject of papers submitted or being submitted (Yohan et al., 2020; J. Mink, in prep, 2022; B. Saintilan, 2021, Cantrell and Michoud, submitted, 2022).

## 6. Conclusion

A 16-year datasets of mapped air pollution concentrations and indicators over France was constructed using a data fusion technique (kriging) that combines measurement from background surface monitoring station and modelling from the regional model CHIMERE. The resulting data are hourly concentrations at a resolution of about 4km over France for the period 2000-2015 (shorter period for $PM_{2.5}$ and $PM_{10}$ hourly indicators).

The kriging technique implemented combines kriging with external drift for $NO_2$ and $O_3$ and co-kriging with external drift for particulate matter, allowing the $PM_{2.5}$ estimation to benefit from the highest density of $PM_{10}$ monitoring stations. These datasets have been evaluated over several years using a cross-validation process that takes into account the incorporation of measurements in the correction process by retaining a data point before calculating the score. The kriging technique

significantly improves the validation scores, especially in urban areas with very low biases and high correlations. However, a point of vigilance appears concerning the representativeness of NO2 concentrations in rural areas which are overestimated by the model. A new methodology is being developed to better map NO2 concentrations in these rural areas. It should be noted that the performance increases with the number of measurements taken into account until a threshold is reached at which the addition of stations no longer seems to improve performance. This threshold dependents on the pollutant, higher for pollutant with a strong spatial gradient (i.e $NO_2$ which has a shorter lifetime).

The main annual indicators (mean $NO_2$, $PM_{10}$, $PM_{2.5}$, $O_3$, SOMO35 and AOT40) are analysed in the document, and annual trends calculated. Significative downward trends are calculated over the whole period for annual average concentrations of $PM_{10}$, $PM_{2.5}$ and $NO_2$. They reflect the reductions in precursor emissions that have taken place in Europe since the 1990s. The trends for $O_3$ over the 16 years are less significant. In general, background $O_3$ level is increasing, mainly due to large-scale pollution and high (peaks) $O_3$ levels are decreasing due to reductions in local $O_3$ precursors emissions. This results in a positive trend for the annual average $O_3$ concentration over most of France, but a small downward trend is also observed in the regions with the highest $O_3$ levels (south-east and east). No significant trend is calculated for the two $O_3$ indicators detailed here (SOMO35 and AOT40). Population exposure is also calculated over France. The average weight of $NO_2$ and $PM_{2.5}$ in the population of the country decreases respectively by 30 % in 16 years and 31 % in 7 years. No clear trend was found for the population weigh of SOMO35.

**Author contribution**

Data kriging, results evaluation by cross-validation process and maps and graph production for the papers were performed by E. Real. The CHIMERE modelling concentration data over the period 2000-2015 were produced by F. Couvidat. Software developments for the kriging and cross-validation methods were provided by A. Ung, L. Malherbe and A. Gressent. The web-based map library used to store and visualised the data has been developed by B. Raux. The all work has been supervised and conceptualized by A. Colette. The manuscript draft has been mainly written by E. Real with contribution of all co-authors.

**Competing interests**

The authors declare that they have no conflict of interest.

**Acknowledgements:**

This work was supported by the French Ministry in Charge of Ecology. Modelling concentration data from the XENAIR program. Part of the simulations were carried out in the XENAIR project funded by the ARC Foundation for Cancer Research.

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
