# Peer review of "Historical reconstruction of background air pollution over France for 2000-2015"

_Earth System Science Data, 2021_

## Author Comment (AC1)

Earth Syst. Sci. Data Discuss., referee comment RC1
https://doi.org/10.5194/essd-2021-182-RC1, 2021
**Comment on essd-2021-182**

Anonymous Referee #1

Referee comment on "Historical reconstruction of background air pollution over France
for
2000–2015" by Elsa Real et al., Earth Syst. Sci. Data Discuss.,
https://doi.org/10.5194/essd-2021-182-RC1, 2021

*Historical reconstruction of air pollution is important for understanding the long-term trends of air pollution and is useful for health studies of air pollution. This paper reconstructs the background air pollution over France for 2000-2015. This work is important. However, the major issue of the paper is lack of novelty. And the methodology used in this paper has not been compared with other models. Besides, the manuscript appears messy a little bit. I cannot stand the terrible typesetting.*

The ambition of our paper is to present and document a new dataset. We believe that we implemented the most up to date and robust methodology, but we do not claim any novelty in producing such a type of historical reconstruction of air pollution. There has indeed been an earlier study on historical reconstruction of outdoor air pollution in France (Bentayeb et al., 2014). But their dataset is not public, which is precisely the gap we are trying to fill here by providing open and transparent access to air pollution exposure data for follow-up studies.

This is the first time that data on pollutant concentrations over France have been made available to the public at this resolution and over such a long time period. This dataset is made available under open access license since July 2020. We are already in close contact with 20 different scientific teams using extensively the dataset. The field of expertise of those teams ranges from epidemiology, to environmental economics and atmospheric science. In order to provide a solid basis for such downstram studies, it is very important that the dataset is clearly documented.
This is why we decided to write this paper and submit it in a journal whose primary aim is to make high quality data available and supported by thorough scientific description.

Finally, regarding the "terrible typesetting", we can only apologies and explain to the reviewer that it is not always obvious for non-native English speakers to develop scientific expertise together with English fluency. We have completely revised the paper trying to improve the English and we hope this will not be regarded as a limitation for the editor and confirm that we would be willing to pay an additional fee for final editing if such a service would be offered by ESSD.

*This paper used a kriging method. Currently, there are many cutting-edge statistical models used for historical reconstruction of air pollution, including many machine learning algorithms. From the results of this paper, the performance of the kriging method is not satisfactory (except for O3). I recommend the authors compare different models and select a model which performs best.*

As a result of the reviewer's suggestion, we compared our cross-validation scores to those found in the literature for air pollution studies in Europe. The following discussion was added p16:

The cross-validation scores can be compared with those obtained in Europe with other mapping methods. Chein et al. (2018) compared 16 algorithms to develop Europe-wide spatial models of PM2.5 and NO2, included linear stepwise regression, regularization techniques and machine learning methods. Those models were developed based on the 2010 routine monitoring data from the AIRBASE dataset, satellite observations, dispersion model estimates and land use variables as predictors. De Hoogh et al.

(2018) also performed cross validation of their fine spatial scale land use regression models (also based on AIRBASE dataset, satellite observations, dispersion model estimates and land use variables as predictors) used in Europe for the year 2010. Results from their cross-validation are compared to our own cross-validation results (without distinction of station type) in Table 13.

Table 13: Validation scores for De Hoogh et al. (2018), Chein et al. (2019) and this study (Real et al. (2022)). The following scores are calculated by cross validation for the 3 studies: Pearson correlation coefficient R2, the bias, and the Root Mean Square Error (RMSE).

| | | De Hoogh et al., 2018 | Chein et al., 2019 | Real et al, 2022 |
|---|---|---|---|---|
| NO2 | R2 | 0.57 | 0.57 - 0.62 | 0.81 |
| | RMSE | 9.51 | 9 - 9.5 | 10.41 |
| | Bias | | | -0.51 |
| PM2.5 | R2 | 0.58 - 0.68 | 0.48 - 0.63 | 0.87 |
| | RMSE | 2.97 - 3.3 | 3.1 - 3.9 | 5.83 |
| | Bias | | | -0.15 |
| O3 | R2 | 0.63 | | 0.92 |
| | RMSE | 6.87 | | 12.54 |
| | Bias | | | -0.07 |

The comparison of performance in these three studies is of course limited by the fact that the spatial coverage differs: in De Hoogh et al. (2018) and Chein et al. (2019), the cross validation is computed over the whole of Europe. In this study, the performances are assessed over France.

For all pollutants the spatial correlation (R2) is better in our study. In the same time, higher RMSE are also found for our study. This may be due to a larger bias, but we also demonstrated in our paper that the bias was very small, except at rural NO2 stations. Snce the RMSE score also depends on the absolute concentrations, the different spatial coverage may also play a role. The lower RMSE over Europe could be an artifact of including areas where absolute concentrations of NO2, PM2.5 or O3 are lower than over France.

The validation scores obtained, as well as the comparison with raw data and with other mapping method, allow us to be confident about the validity of the concentrations obtained and their good representativeness of background concentrations, in particular in urban areas. A point of vigilance appears however when it comes to the representativeness of rural NO2 concentrations which are overestimated in our results.

I don't think it is appropriate to add a reference in the abstract (i.e., Real et al., 2021). (Line 18, P1)

This is required by the editor

There have been many studies about the historical reconstruction of air pollution. They should conduct a thorough literature review in the introduction section.

This following discussion was added L3P3 of the original manuscript:

"These choices are the results of successive studies that compared different kriging techniques (Malherbe and Ung, 2009, Beauchamp 2015a). A similar methodology was implemented for an earlier reconstruction of outdoor air pollution in Europe for the period 1989-2008 in (Bentayeb et al., 2014).

There are also ambient air pollution maps produced at European scale at 1km resolution by the European Environment Agency, but only for selected annual indicators and without consistency for multi-year reconstructions (Horálek et al., 2012, 2020). The Copernicus Atmosphere Monitoring Service has also produced European analyses since 2015, but again there is no multi-year consistency as these European maps are produced on an annual basis with gradually improving methodologies (Marécal et al., 2015). At Global scale, the Global Burden of Disease also makes available air pollution exposure maps, a recent update of the methodology was presented in (Shaddick et al., 2017), but the resolution is 0.1 degrees or about 10km."

Line 26, P1 – Line 13, P2: It is not necessary to describe the trends of air pollution trends coming from ground observations in detail in the Introduction section. These contents have little to do with the purpose of the historical reconstruction of air pollution in France. These contents can be moved to the Results and Discussion section. They can compare their results of the trends using the reconstruction data and the results from previous studies using ground observations.

We have reduced this part in the introduction and introduced elements of comparison in the different sections referring to the trend analysis.

Exemple 1:Section 4.1.1, p21: "Taking the year 2000 as the base year, this amounts to a 39% reduction. In a study conducted for France over the period 2000-2010, Malherbe et al. (2017) estimated a downward trend that was twice as small (0.4). This reflects the accelerated decline in concentrations in France in recent years."

Exemple 2:

Section 4.1.4, p29: "This downward trend is slightly stronger than that calculated in Malherbe et al. (2017) over the period 2000-2010 over France (-0.37 $\mu$g.m$^{-3}$.year$^{-1}$) and corresponds to a reduction of about 30% (taking 2020 as the base year)."

Line 22, P3. They exclude industrial and traffic stations. In this case, the reconstruction maps of air pollutants will miss many pollution hot spots. I know that they want to reconstruct the background air pollution. However, without these hot spots, the reconstruction of air pollution is not that useful. I think another reason they exclude these stations is that the kriging method cannot deal with these stations with higher pollution levels well, because these stations are much less than urban and rural stations. However, the machine learning algorithms with land use information as covariates can capture the high pollution hot spots. Of course, they also need to incorporate meteorological variables in the models.

As stated in the paper, and as specified by the reviewer, the data proposed here are intended to reproduce background concentrations in France. Given the resolution of our data (about 4km2), the simulated concentrations on a grid cell must be representative of the average of the real concentrations on this grid cell. However, traffic and industrial stations are representative of more local concentrations, which evolve rapidly when air masses move away from these sources. For the resolution proposed in this paper, it is therefore a sensible choice to use only background stations, which are further away from the sources and therefore more representative of the concentrations at the scale of the grid cell. It would of course be interesting to go down in resolution and propose maps on such time scales at the sub-kilometer scale. In this case, the addition of traffic and industrial stations would be justified. We have modified our discussion/conclusion in this sense.

Line 22, P3. Why the number of the stations of PM2.5 in 2007 are much fewer than that in 2006 and 2008?

We exchanged with our colleagues who are specialists in particle measurements. Before 2009, the data are based on a mix of TEOM and TEM-FDMS data, but with very few reference measurements (FDMS). In view of the discontinuity and low reliability of the data before 2009, they advised us not to use these data. This was already the case, but we have revised the text and the table accordingly:

Section 2.1: "Concerning PM2.5, given the few reference measurements available before 2009, the reliability of even annual measurements is low. It was therefore decided to apply the kriging methodology only from the year 2009 onwards, for which the change in measurement method had become widespread. ».

The number of PM2.5 stations is shown from the year 2009 onwards now.

Line 14, P5. Move "(particles with a radius < 10 μm)" and "(particles with a radius < 2.5 μm)" to the places where PM2.5 and PM10 first appear.

OK

P6, "3. Data validation". I think the leave-one-station-out CV cannot capture the model overfitting issues well. Typically, 10-fold spatially CV (leave-10%-station-out CV) is commonly used in such kind of studies.

We added the following sentences at the beginning of section 3:

"Leave-one-out validation is a commonly used method in the air quality community (see for example ETC reports on air quality mapping (ETC, 2020)) which is presently recommended by FAIRMODE (FAIRMODE guidance, 2020). However scores derived from the results of the leave-one-out validation might be influenced by areas where the density of sampling points is highest. For this reason, during the FAIRMODE project (Riviere et al., 2019), for which a kriging method similar to the one conducted here was conducted, a comparison has been performed between cross-validation results obtained by the leave-one-out cross-validation and cross-validation results obtained by the 5-fold cross validation (leave-20%-station-out CV). Results and related scores were very similar. We therefore decided to keep to the leave-one-out cross-validation process for the validation of our kriging results."

The chapter and section numbers are messy: " Data validation"->" 3.1.4. PM10"->" 3.1.5. PM2.5"->" 3.1.6. O3"->" 3.1.7. NO2"->" 4. Results"->" 4.1 Concentration maps and trends"-> "3.1.1. PM10"->" 3.1.2. PM2.5"->" 3.1.3. Ozone"->" 3.1.8. NO2"->" 4.2 Exposure trends"->" 4. Data availability".

OK, this has been corrected

The words in the figure are too small. (e.g., Figure 9, etc.)

Figures 9 to 15 have been enlarged

Change "4. Results" to " 4. Results and Discussion"

OK

Incorporate the section "4 Data availability" into "Conclusion" section.

This section is required by the Editor

The figures and tables can be better-looking.
We have enlarged the figure 9 to 15 to make them more readable

---

## Author Comment (AC2)

Earth Syst. Sci. Data Discuss., referee comment RC2
https://doi.org/10.5194/essd-2021-182-RC2, 2021
Comment on essd-2021-182

Anonymous Referee #2

Referee comment on "Historical reconstruction of background air pollution over France for 2000–2015" by Elsa Real et al., Earth Syst. Sci. Data Discuss., https://doi.org/10.5194/essd-2021-182-RC2, 2021

The paper by Real et al. proposes a dataset of background air pollution concentrations and air quality indicators over France for the period 2000-2015. The concentrations and indicators are mainly given on an annual basis either gridded at about 4km resolution or aggregated on French administrative territories. The provided concentrations are calculated using kriging approaches merging surface measurements from air quality networks and model simulation. The evaluation of the dataset is done using a crossvalidation method and shows good performances of the dataset to assess air pollution concentrations except for NO2 at rural stations. Trends of the different pollutants (PM10, PM2.5, O3, and NO2) are discussed as well as exposure trends. The dataset covering the 2000-2015 period is available on a zenodo repository. In addition, the visualization of the maps is also available on the INERIS website with the possibility to download the data for more recent years. The presented dataset is of interest for air quality community, for example for comparison of air pollution trends in different countries, the dataset providing information for France. It is then suitable for publication, but some major issues should be addressed before publication (see point 1 and 2 of main comments):

Main comments:
The description of the kriging approaches is very limited in the paper and most of the references provided by the authors are written in French, limiting the access to non-French speaking readers. Providing a more detailed summary of the approaches would be valuable for the readers. The presented dataset is a fusion between model simulations and surface measurements.

We have given a more precise and detailed description of the kriging method used in this paper, as well as more extensive international references (see paragraph 2.3):

[revised manuscript text omitted]

The authors do not provide any evaluation or discussion of the improvements provided by the kriging approaches compared to the raw model simulations. It would be very valuable to have this information to highlight the usefulness of the dataset compared to raw simulations. Is it possible to calculate the contribution of the model vs surface measurements for each grid point?

We calculated the validation scores for the raw data and added the following text to the paper on p. 16 (new section: 3.5: Comparisons with other scores):
" In order to evaluate the added value of the kriging technique compared to the raw CHIMERE model simulations, the cross-validation scores can be compared to the raw model scores. Table 1 shows the scores averaged over all years and all background observations, without distinction of typology.

**Table 1: Validation scores for the raw data and the kriged concentrations (cross-validation). Annual scores (bias, RMSE and the Pearson correlation coefficient $r^2$) are calculated over France for all year and all stations and are averaged.**

|  | $NO_2$ | $O_3$ | $PM_{10}$ | $PM_{2.5}$ |
|---|---|---|---|---|
| RAW | | | | |
| Bias | -3.51 | 3.46 | -8.91 | -4.02 |
| RMSE | 12.97 | 17.26 | 12.63 | 8.73 |
| $R^2$ | 0.55 | 0.73 | 0.71 | 0.75 |
| KRIGED CONCENTRATION | | | | |
| Bias | -0.51 | -0.07 | -0.04 | -0.15 |
| RMSE | 10.41 | 12.54 | 7.64 | 5.83 |
| $R^2$ | 0.81 | 0.92 | 0.85 | 0.87 |

All scores are clearly improved by the kriging of observations using CHIMERE as external drift. However, as can be seen in the previous figures, this improvement is more pronounced in urban areas than in rural areas, due to the much larger number of stations in urban areas, which makes the kriging more representative of these areas. "

The authors discussed the significance of the trends at the national scale, but few information is given when trend maps are presented. Are the trends significant at each grid point?

The representative confidence interval maps have not been included in the paper to avoid cluttering it up, but discussions of their results have been added.
Ex:
P22 (section 4.1): This trend is statistically significant on average over France with a narrow 95%-confidence interval ([-0.50;-1.09]) that does not include zero (see **Erreur ! Source du renvoi introuvable.**) and applies to almost all grid points (maps of confidence interval, not shown here)
p28 (4.1.3): When considering ozone, however, according to the value of the mapped 95 % confidence interval (not shown here) on most grid points, the confidence interval is wide and contains zero, indicating a lack of significance of the calculated trends.

A proofreading by a native English speaker is recommended.

Specific comments:
Page 2, lines 2-5: the authors should refer to the Tropospheric Ozone Assessment Report (TOAR activity from IGAC) when discussing tropospheric ozone trends.

The following reference has been added:

Tarasick, D., Galbally, I.E., Cooper, O.R., Schultz, M.G., Ancellet, G., Leblanc, T., Wallington, T.J., Ziemke, J., Liu, X., Steinbacher, M., Staehelin, J., Vigouroux, C., Hannigan, J.W., García, O., Foret, G., Zanis, P., Weatherhead, E., Petropavlovskikh, I., Worden, H., Osman, M., Liu, J., Chang, K.-L., Gaudel, A., Lin, M., Granados-Muñoz, M., Thompson, A.M., Oltmans, S.J., Cuesta, J., Dufour, G., Thouret, V., Hassler, B., Trickl, T. and Neu, J.L., Tropospheric Ozone Assessment Report: Tropospheric ozone from 1877 to 2016, observed levels, trends and uncertainties. Elem Sci Anth, 7(1), p.39. DOI : 10.1525/elementa.376, 2019

Page 4, CHIMERE description: the meteorological fields used as input of model simulations are different depending on the period (WRF from 2000 to 2010 and IFS from 2011). Does the change of systems to constrain the meteorological fields introduce any bias or discontinuity in the simulations?

It is indeed possible that the change in meteorological data between the period 2000-2010 and 2010-2015 has led to changes in the raw data. The evolution of the comparison scores of the raw model with the observation data seems to show higher correlations ($r^2$) after 2010 (not shown in the paper). However, it is difficult to know whether this can be attributed to meteorology alone since the emissions are also different. Furthermore, the WRF simulations themselves where nudged within ECMWF reanalyses, so they are not independent from IFS setup. Lastly, the data we produced are adjusted data using kriging methods. The impact of using either of the meteorological data sources will therefore be offset by the data fusion technique.

Figure 1 and similar: the dashed lines are confusing; they may be interpreted as error bars. They are not commented in the caption.
We added a description of those dashed lines in the figure captions.

Please check the size of the text in figures, it is sometimes too small, especially for figures from fig. 9.

Figures 9 to 15 have been enlarged.

Figure 9: the term "reanalysis" is used in the figure but never used in the text. Please use consistent terms all over the paper or define them clear (kriging, fusion, reanalysis).

The text has been made more consistent.

---

## Author Response (AR2)

**Suggestions for revision or reasons for rejection (will be published if the paper is accepted for final publication)**

I am glad that the authors has revised the manuscript according to the reviewer's comments.

They said that this dataset has been used in some epidemiological studies. Is there any study published? I am interested in it. If possible, they could provide some examples using this dataset, especially those have been published.

We contacted all the researchers to whom we had provided the data. Most of the work undertaken with our data is still in progress, but others are the subject of papers submitted or in the process of being submitted. We have therefore added the following paragraph to the section 5; Data availability:

*"Those data have been provided to several research teams with different field of expertise ranging from epidemiology, to environmental economics and atmospheric science. Most of this work is still in progress, but others are the subject of papers submitted or being submitted (Yohan et al., 2020; J. Mink, in prep, 2022; B. Saintilan, 2021, Cantrell and Michoud, submitted, 2022)."*